# Towards Interpretable Visual Decoding with Attention to Brain Representations

**Pinyuan Feng**[1]* **Hossein Adeli**[1] **Wenxuan Guo**[1]

**Fan Cheng**[1] **Ethan Hwang**[1] **Nikolaus Kriegeskorte**[1]

[1]Zuckerman Mind Brain Behavior Institute, Columbia University, USA

🌐 Project Page

## Abstract

Recent work has demonstrated that complex visual stimuli can be decoded from human brain activity using deep generative models, offering new ways to probe how the brain represents real-world scenes. However, many existing approaches first map brain signals into intermediate image or text feature spaces before guiding the generative process, which obscures the contributions of different brain areas to the final reconstruction output. In this work, we propose *NeuroAdapter*, a visual decoding framework that directly conditions a latent diffusion model on brain representations, bypassing the need for intermediate feature spaces. Our method demonstrates competitive visual reconstruction quality on public fMRI datasets compared to prior work, while providing greater transparency into how brain signals drive visual reconstruction. To this end, we introduce an Image–Brain BI-directional interpretability framework (*IBBI*) that analyzes cross-attention patterns across diffusion denoising steps to reveal how different cortical areas influence the unfolding generative trajectory. Our work highlights the potential of end-to-end brain-to-image reconstruction and establishes a path for interpretable neural decoding.

## 1 Introduction

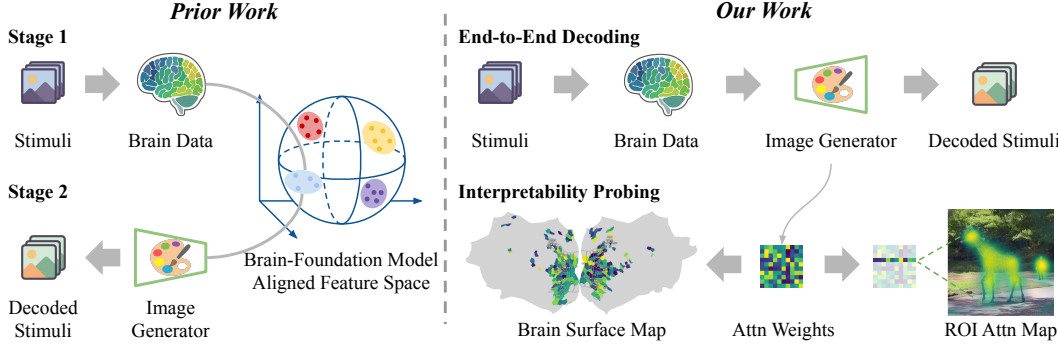

Figure 1: **Overview**. *Left*: Typical two-stage decoding pipelines first map brain activity to intermediate feature spaces (e.g., CLIP/DINO) and then use those embeddings to guide a generative model. *Right*: Our end-to-end approach conditions a latent diffusion model directly on brain activity, enabling interpretations of the generative dynamics in both image and brain spaces.

Understanding how the human brain represents the visual world remains a central challenge in neuroscience. Neural decoding approaches help address this challenge by inferring the content of the

---

*pf2477@columbia.edu

representation in different brain areas – or across the whole brain – in response to complex stimuli. In recent years, decoding models have achieved remarkable success across different perceptual modalities and intended movements, with many pipelines incorporating deep generative models. These works have pushed the NeuroAI frontier of reconstructing content or decoding "thoughts" from brain activity, bringing the prospect of "mind reading" closer to reality.

Current approaches to reconstructing visual stimuli from brain activity (Lin et al., 2022; Cheng et al., 2023; Takagi & Nishimoto, 2023; Ozcelik & VanRullen, 2023; Scotti et al., 2023; Li et al., 2025; Ferrante et al., 2025) typically implement a two-stage pipeline (Fig. 1, left): (i) brain activity is first mapped to intermediate image- or text- embeddings derived from large foundation models (e.g. CLIP (Radford et al., 2021) and DINO (Caron et al., 2021; Oquab et al., 2023; Siméoni et al., 2025)); (ii) these intermediate representations are then used to condition a visual generative model for stimulus reconstruction. Mapping brain data into an intermediate representation space leverages rich priors in embedding spaces to improve reconstruction quality and has proved highly effective for reconstruction. However, the use of this intermediate representation can introduce an information bottleneck (Mayo et al., 2024; Shirakawa et al., 2025), with successful reconstruction of perceived stimuli depending on the alignment between neural representations and the embedding space. This intermediate step can also mask the effect of different brain areas on the final reconstruction, limiting the interpretability of the approach. In this work, we explore an alternative approach (Fig. 1, right) to two-stage decoding pipelines: conditioning latent diffusion models directly on the brain activity.

**Contributions of our paper.** Our contributions are as follows: (1) we propose *NeuroAdapter*, an end-to-end framework that learns parcel-wise embeddings from fMRI data and integrates them into latent diffusion models through cross-attention; (2) we show that our approach achieves competitive performance on public fMRI datasets, demonstrating that high-quality visual reconstructions can be obtained without reliance on external embedding spaces; and (3) we provide a bi-directional interpretability framework, namely *IBBI*, which leverages cross-attention dynamics across diffusion steps to reveal both the relative contribution of brain parcels and their spatial influence in the reconstructed images, offering new insights into the generative process from a neuroscientific perspective.

## 2 RELATED WORK

**Brain Decoding with Deep Generative Models.** Early pioneering work demonstrated that fMRI signals could be decoded into continuous visual experiences by treating reconstruction as a stimulus identification task. For example, Nishimoto et al. (2011) used a motion-energy encoding model and Bayesian inference to retrieve viewed movie clips from a large library of candidates. With the rise of deep generative modeling, decoding has progressed from classification to photorealistic reconstructions that leverage powerful image priors. Early GAN-based pipelines established the feasibility of mapping brain signals into deep feature spaces and synthesizing images (Seeliger et al., 2018; Shen et al., 2019a;b; Cheng et al., 2023; Gu et al., 2024). Latent diffusion has since become the dominant image prior, with several methods steering Stable Diffusion via fMRI-predicted image/text latents (Lin et al., 2022; Chen et al., 2023; Ozcelik & VanRullen, 2023; Scotti et al., 2023; Takagi & Nishimoto, 2023; Zeng et al., 2024; Wang et al., 2024b).

Recent studies have experimented with different conditioning inputs, training regimes, or cross-subject alignment strategies (Xia et al., 2024; Han et al., 2024; Huo et al., 2024; Li et al., 2025; Wang et al., 2024a; Gong et al., 2025). In particular, Ferrante et al. (2024) has shown that aligning NSD subjects' fMRI into a shared functional space enables cross-subject reconstruction. Despite this progress, most pipelines still route brain activity through intermediate vision or vision-language feature bottlenecks to guide generations. The latest streamlined approach, Dynadiff (Careil et al., 2025), moved towards a single-stage solution by using LoRA finetuning (Hu et al., 2022) for dynamic visual decoding from time-resolved fMRI signals. In contrast, our proposed *NeuroAdapter* conditions the latent diffusion model directly on brain representations via cross-attention, enabling a more transparent and anatomically grounded interface between fMRI signals and the generative model.

**Interpretable Visual Decoding.** A central goal of visual neuroscience is to understand both the *functional selectivity* of brain areas (what information they encode) and the *representational format* of that information. *Encoding* approaches advance the first goal by learning a brain encoder that

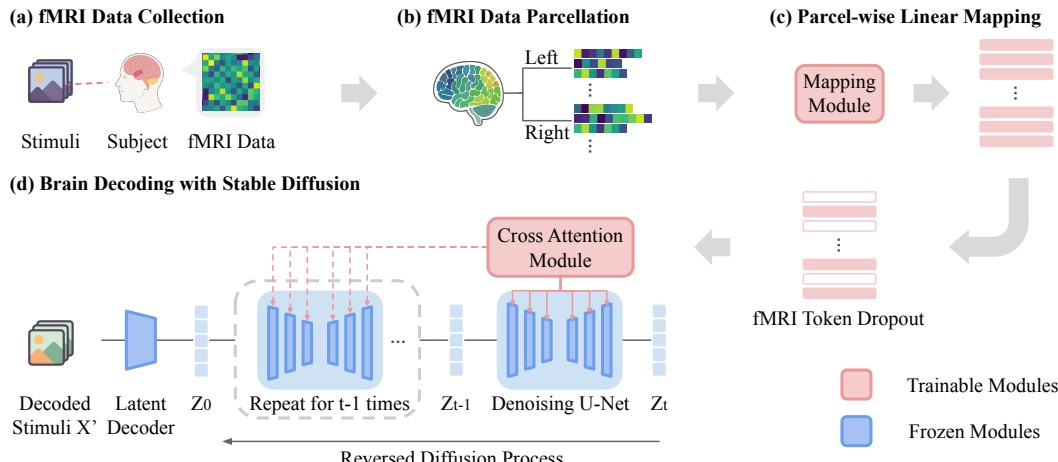

Figure 2: **NeuroAdapter training pipeline.** (a) fMRI data collection paradigm, (b) cortical parcellation, (c) parcel-wise linear mapping from vertices to brain representation tokens, and (d) conditioning a latent diffusion model on these tokens for reconstruction.

maps images to neural activity, and then using this encoder to (i) optimize stimuli that maximally drive a given cortical region (Luo et al., 2023a) or (ii) generate natural-language descriptions of voxel-level selectivity (Luo et al., 2023b). Complementarily, transformer-based brain encoders provide an interpretable architecture whose attention maps explicitly route visual features into distinct brain areas (Adeli et al., 2025), offering mechanistic insight into functional organization (Hwang et al., 2025). In contrast, *decoding* approaches target the second goal by testing what can be *read out* from neural activity and how reconstructions depend on specific regions, thereby probing the format and distribution of visual information. Studies that train and test decoders on subsets of visual areas have revealed how information is distributed across the visual hierarchy (Shen et al., 2019a;b; Horikawa & Kamitani, 2022; Cheng et al., 2023; Ozcelik & VanRullen, 2023). Parallel developments in language neuroscience introduce interpretable embeddings and causal testing frameworks to link representational dimensions to brain activity (Tang et al., 2023a; Benara et al., 2024; Antonello et al., 2024).

A key scientific motivation for decoding, alongside encoding analyses, is that they address complementary questions. Encoding models characterize how external stimuli are transformed into neural responses. Decoding instead asks what aspects of visual or mental content can be reliably read out from measured neural activity, which is particularly important in settings where the relevant subjective percept is only partially constrained or cannot be fully specified by an external stimulus, e.g., visual illusion (Cheng et al., 2023), mental imagery (Kneeland et al., 2025), dreams (Horikawa et al., 2013), and other forms of subjective perception. To leverage decoding for scientific insight, it is essential to understand how a decoding model uses brain signals to guide image generation. Existing analyses of latent diffusion models examine when (diffusion time step) and where (model layer) low- and high-level features emerge in the model (Takagi & Nishimoto, 2023), but typically lack a dynamic view of which parts of the generated image are modulated by brain-derived information. Our work addresses this gap by using cross-attention to provide explicit, temporal maps linking brain signals to image regions throughout the denoising process.

## 3 METHODS: MODEL TRAINING AND EVALUATION

Our brain decoding model, *NeuroAdapter*, as shown in Fig. 2, is built on the IP-Adapter framework (Ye et al., 2023). We conditioned a pre-trained Stable Diffusion model [1] (SD; Rombach et al. (2022)) on fMRI-derived features via cross-attention mechanism to reconstruct perceived visual stimuli. In this section, we explain the details of our method with the Natural Scene Dataset (NSD; Allen et al. (2022)), but a similar method applies to the other datasets as well.

---

[1]https://huggingface.co/stable-diffusion-v1-5/stable-diffusion-v1-5

## 3.1 Neural Data Processing and Parcellation

We trained our model using the surface-based fMRI data in *fsaverage* space. We first averaged the vertex responses across image repetitions to obtain a single response pattern per image. To transform the high-dimensional fMRI data into structured inputs for conditioning the diffusion model, we applied the Schaefer parcellation ((Schaefer et al., 2017); see Appendix C). This clusters cortical vertices into 500 parcels per hemisphere and has been shown to be an effective practice for brain tokenization (Bosch et al., 2025).

To improve robustness of the model by restricting inputs to high-quality regions, we computed vertex-wise Signal-to-Noise Ratio (SNR) and selected top $k$ parcels per hemisphere with the highest average SNR, yielding a total of $p = 2k$ parcels as fMRI conditioning inputs to the model. In the following sections, we report results of our model trained on $p = 200$ brain parcels, and present an ablation study on how varying $p$ influences decoding performance in the Appendix I.

## 3.2 Parcel-wise Linear Mapping

Since the number of vertices varies across parcels, we padded each parcel's vertex response vector to match the largest vertex count across parcels $v_{max}$. This yields processed neural data $D_{fMRI} \in \mathbb{R}^{n \times p \times v_{max}}$, where $n$ is the batch size of stimulus images. Then, each parcel was assigned a unique projection matrix $w \in \mathbb{R}^{v_{max} \times f}$, transforming padded vertex response into fMRI token embeddings $E \in \mathbb{R}^{n \times p \times f}$, where $f$ is the hidden dimension of fMRI token embeddings. In the main text, we set $f = 768$ during model training, and results from an ablation study with different values of $f$ is provided in the Appendix H. Additionally, we conducted another ablation study (Appendix J) to demonstrate that mapping fMRI data into the parcel-wise token space to condition the SD generation is effective for visual reconstruction.

## 3.3 Latent Diffusion Process with Brain Conditioning

We replaced the cross-attention layer of the U-Net (Ronneberger et al., 2015) in SD with an IP-adapter-style cross-attention module (Ye et al., 2023), enabling the model to attend to the fMRI token embeddings. To ensure that embeddings were the only conditioning input, the text encoder in SD received an empty input during both training and inference. During training, only the parcel-wise linear mapper and the new cross-attention modules were updated, with the rest of the parameters kept frozen.

**fMRI Token Dropout.** We applied a stochastic token dropout strategy during training to the fMRI token embeddings $E$ to ensure robustness of visual decoding. We randomly dropped out parcel-wise token vectors for each training sample. A dropout probability $r \sim \mathcal{U}(0, 1)$ was drawn, and each fMRI token vector was independently retained with probability $1 - r$. This resulted in a binary mask $M \in \{0, 1\}^{n \times p \times 1}$, which was applied parcel-wise to the fMRI token embeddings $E' = E \odot M$. We found this regularization to be crucial for strong decoding performance, as supported by the ablation results in Appendix G.

**Min-SNR Loss Weighting.** To stabilize training and improve sample quality, we adopted the min-SNR weighting strategy (Hang et al., 2023) recently introduced in diffusion models. This approach down-weights the contribution of easy high-SNR steps, where reconstructions are clean, while preserving the weight of noisy low-SNR steps, yielding a more balanced training signal across the diffusion process (please view Appendix M for details).

## 3.4 Decoded Image Selection with Brain Encoding Model

Inspired by Kneeland et al. (2023), we used a whole-brain encoder (Adeli et al., 2023; 2025; Hwang et al., 2025) trained on the same fMRI-image training dataset to identify the best decoded stimuli during evaluation. As shown in Fig. 3 (a), for each fMRI sample in the test set, the decoder generated a set of candidate images $X'_0, \cdot, X'_n$ with $n$ different random seeds. The brain encoder predicted vertex-wise fMRI activity $B'_0, \cdots, B'_n$ for the candidate images, which was correlated with the ground-truth fMRI measurements. The candidate image with the highest Pearson correlation was

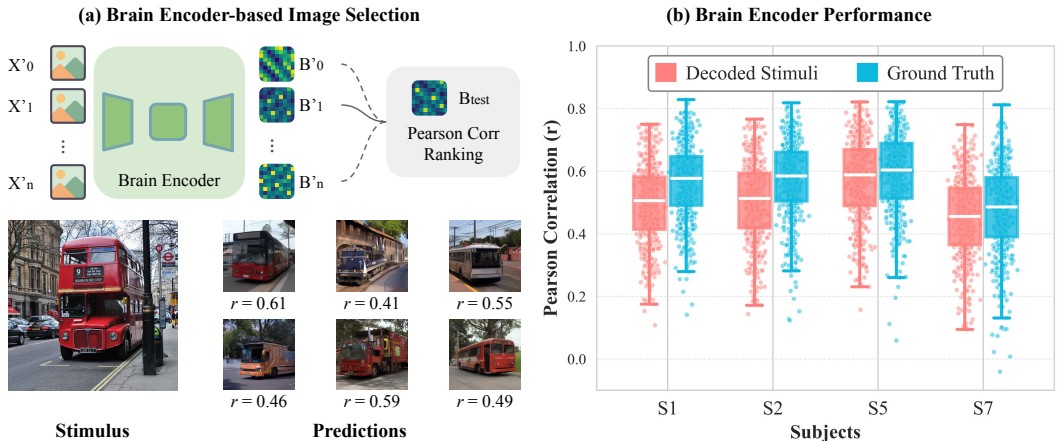

Figure 3: **Brain Encoder.** (a) Brain encoder–based image selection using Pearson correlations between predicted and measured fMRI responses for an NSD test example. (b) Red: correlation between the predicted brain activity from the decoded images and the measured brain activity. Blue: correlation between the predicted activity for the stimulus in testing set and the corresponding fMRI response.

selected as the final decoded image for further evaluation. An ablation study assessing the impact of the brain encoder to decoding performance is reported in Appendix K.

## 4 METHODS: IBBI FRAMEWORK FOR INTERPRETABILITY

Beyond decoding performance, we also investigated the interpretability of the generative process in our model. During inference, the SD model reconstructs images by progressively denoising a latent representation over multiple steps, starting from pure Gaussian noise and gradually refining it toward a clean image. At each denoising step $t$, the U-Net backbone applies a sequence of downsampling and upsampling blocks, each equipped with cross-attention layers that integrate the fMRI-derived conditioning. Since the conditioning input to SD was parcel-wise embeddings, this can be represented as a token matrix $E \in \mathbb{R}^{p \times f}$ (batch size $n = 1$ for simplicity), where each row $e_i \in \mathbb{R}^f$ corresponds to the embedding of parcel $P_i$. If anatomical or functional labels are available for brain parcels, this formulation enables ROI-level probing of the cross-attention mechanism to see how brain representations interact with U-Net in the generative process. Following this idea, we propose the **I**mage-**B**rain **BI**-directional framework (*IBBI*) for exploring the internal attention dynamics, which links brain activity and image features during decoding.

### 4.1 PROBLEM SETUP

In *NeuroAdapter*, each cross-attention layer computes attention scores $\text{Attn}(Q, K, V)$, where queries $Q \in \mathbb{R}^{q \times d}$ come from spatial tokens in the U-Net of SD, and keys and values $(K, V) \in \mathbb{R}^{p \times d}$ are derived from the fMRI embeddings $E$. At each denoising timestep $t$, the attention weight matrix $A^{(\ell, h, t)} \in \mathbb{R}^{q \times p}$ for head $h$ in layer $\ell$ encodes the influence of each parcel token on each spatial query. Each entry of the attention weight matrix can be expressed as:

$$A_{i,j}^{(\ell,h,t)} = \frac{\exp\left(\langle Q_i^{(\ell,h,t)}, K_j^{(\ell,h,t)} \rangle / \sqrt{d}\right)}{\sum_{j'=1}^{p} \exp\left(\langle Q_i^{(\ell,h,t)}, K_{j'}^{(\ell,h,t)} \rangle / \sqrt{d}\right)}$$

where query index $i \in \{1, \ldots, q\}$, and parcel index $j \in \{1, \ldots, p\}$. Specifically, the entry $A_{i,j}^{(\ell,h,t)}$ refers to the attention from the $i$-th query vector $Q_i^{(\ell,h,t)}$ to the $j$-th parcel token, represented by its key vector $K_j^{(\ell,h,t)}$. Intuitively, each entry of this matrix reflects the degree of attention from a particular spatial query in the image to a specific parcel. Our proposed interpretability framework further exploits this matrix from two complementary views.

## 4.2 BRAIN-DIRECTED VIEW

We summarize the attention weight matrix $A^{(\ell,h,t)}$ over brain parcel tokens at each timestep into a vector $B^{(t)}$ (parcel contribution vector), normalized to unit mass. Formally, let $L$ be the number of cross-attention layers in U-Net, $H$ be the number of multi-attention heads, and $q^\ell$ be the number of spatial queries in layer $\ell$, At the denoising step $t$, each cross-attention map satisfies $\sum_{j=1}^{p} A_{i,j}^{(\ell,h,t)} = 1$ for every $(\ell, h, i)$. To aggregate the total attention mass assigned to each parcel across layers with different spatial resolutions, we weight every query equally and normalize by the total number of queries $\sum_{\ell=1}^{L} q^\ell$. For each parcel $j \in \{1, \ldots, p\}$, we define

$$B_j^{(t)} \;=\; \frac{1}{H \sum_{\ell=1}^{L} q^\ell} \sum_{\ell=1}^{L} \sum_{h=1}^{H} \sum_{i=1}^{q^\ell} A_{i,j}^{(\ell,h,t)}$$

Here, $\sum_{j=1}^{p} B_j^{(t)} = 1$, so $B^{(t)} \in \mathbb{R}^p$ can be interpreted as a query-weighted share of attention mass over parcels at timestep $t$. The vector represents the *relative contribution* of different parcels.

## 4.3 IMAGE-DIRECTED VIEW

We are motivated by the previous work that interpreting text guidance in SD (Tang et al., 2023b). In our case, the spatial structure in $A^{(\ell,h,t)}$ enables us to explore further where, in the generated image, each brain parcel or ROI (Region of Interest) directs its attention at timestep $t$. For a given ROI group from parcels, denoted as $\mathcal{R} \subseteq \{1, \ldots, p\}$, we pool attentions across heads and ROI tokens to form a query-wise attention profile for each layer:

$$m_{\mathcal{R}}^{(\ell,t)}(i) \;=\; \frac{1}{H} \frac{1}{|\mathcal{R}|} \sum_{h=1}^{H} \sum_{j \in \mathcal{R}} A_{i,j}^{(\ell,h,t)}$$

The vector $m_{\mathcal{R}}^{(\ell,t)} \in \mathbb{R}^{q^\ell}$ is then reshaped to a 2D map, which matches the spatial grid of the layer $\ell$. Because the spatial resolution varies across downsampling and upsampling blocks of the U-Net, we upsample each 2D map to full image resolution, yielding $U_{\mathcal{R}}^{(\ell,t)} \in \mathbb{R}^{H_{\text{img}} \times W_{\text{img}}}$ for every cross-attention layer. To produce overlays that are comparable for spatial location across ROIs, we normalize each upsampled map to unit $L_1$ mass and then average uniformly across layers:

$$I_{\mathcal{R}}^{(t)} \;=\; \frac{1}{L} \sum_{\ell=1}^{L} \frac{U_{\mathcal{R}}^{(\ell,t)}}{\sum_{x,y} U_{\mathcal{R}}^{(\ell,t)}(x,y)}$$

We refer to $I_{\mathcal{R}}^{(t)}$ as the ROI attention maps, which highlights *where* a given ROI allocates its attention in the image at timestep $t$. Intuitively, ROI attention maps provide a functional footprint of each ROI in the stimulus space, allowing us to interpret the role of neural data from different parts of the brain in shaping specific image regions during reconstruction.

## 5 EXPERIMENTS

### 5.1 DATASETS

**Natural Scene Dataset (NSD).**  We used the NSD, a large-scale 7T-fMRI dataset designed for studying visual representations in the human brain (Allen et al., 2022). This contains high-resolution brain responses from eight subjects, each viewing up to 10,000 distinct natural images sampled from the MSCOCO dataset (Lin et al., 2014). In our experiments, we trained our brain decoding model and encoding model (see Section 3.4) on the NSD data following the standard preprocessing steps. In the following sections, we report comparison with prior work using the averaged results from four subjects who completed all fMRI scanning sessions (subjects 1, 2, 5, 7). For the relevant ablation studies, we restricted our analysis to subject 1 and evaluated models under different experimental conditions on this single-subject dataset.

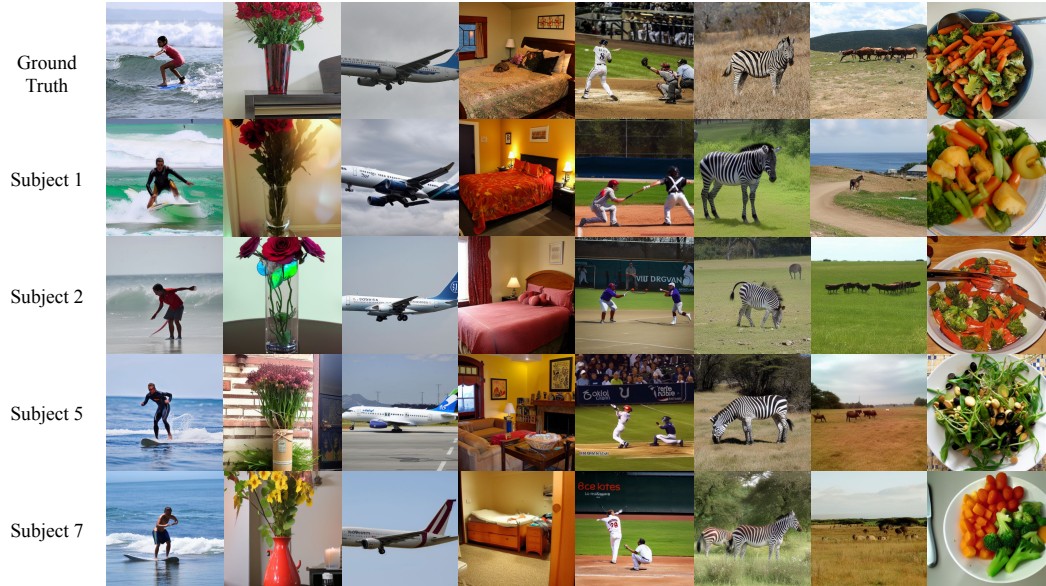

Figure 4: **Ground truth with decoded stimuli from *NeuroAdapter* across 4 subjects.**

**NSD-Imagery.** We further evaluated our framework on the NSD-Imagery dataset (Kneeland et al., 2025), an extension of the NSD designed to study brain activity during mental imagery. It contains high-resolution 7T-fMRI recordings from the same eight participants as NSD, with trials including simple geometric patterns, complex natural scenes, and conceptual word cues. During imagery runs, subjects were cued with a letter and instructed to vividly imagine the corresponding stimulus without physically seeing it. Each subject completed 12 runs (9 run types with imagery runs repeated twice), yielding 576 trials per participant. In evaluation, we directly tested our model, which was trained on NSD, on this dataset to see if our model can generalize to mental imagery tasks.

**Deeprecon Dataset (Deeprecon).** The Deeprecon dataset (Shen et al., 2019b) comprises fMRI activity data from five subjects who viewed both ImageNet images and artificial images. The dataset contains 1,200 distinct natural images for training (each presented with five repetitions), 50 natural images and 40 artificial images for testing (each presented over 20 repetitions), totaling 8,000 brain samples per subject. An important consideration for this dataset was that natural test images were selected from ImageNet categories that differed from the training categories, and artificial images were included as additional test stimuli. For this dataset, we trained our brain decoder and encoder on 16 brain parcels across the two hemispheres, including early visual areas (V1, V2, V3), V4, higher-order visual regions (LOC, FFA, PPA), and the broader higher visual cortex (HVC) region.

## 5.2 EVALUATION METRICS

We evaluated the model's performance using the following eight image quality metrics that are commonly used in the literature. *PixCorr* measures the pixel-level correlation between reconstructed and ground-truth images. *SSIM* denotes the Structural Similarity Index Metric (Wang et al., 2004). *AN(2)* and *AN(5)* refer to the 2-way classification (2WC) accuracy based on features from layers 2 and 5 of AlexNet (Krizhevsky et al., 2012), respectively. *CLIP* corresponds to the 2WC accuracy of the output layer of the ViT-L/14 CLIP-Vision model (Radford et al., 2021). *Incep* refers to the 2WC accuracy computed on the final pooling layer of InceptionV3 (Szegedy et al., 2016). *Eff* and *SwAV* are distance-based metrics computed using feature representations from EfficientNet-B13 (Tan & Le, 2019) and SwAV-ResNet50 (Caron et al., 2020).

## 5.3 DECODING DYNAMICS ANALYSIS VIA CROSS ATTENTION

During inference, we applied 50 denoising steps for the reversed diffusion process, which is a common practice for Stable Diffusion (Appendix L). We extracted the full attention weight matrices

$A^{(\ell,h,t)}$ across all layers $\ell$ at each timestep. This yields a step-by-step record of how brain representations influence different spatial queries throughout the generative trajectory. For brain-directed view, we computed a parcel contribution vector $B^{(t)}$ showing the relative influence of each parcel at each timestep. We then projected this vector onto the cortical surface using `pycortex` (Gao et al., 2015), visualizing how strongly each parcel influenced the generated stage. For image-directed view, we mapped the spatial query tokens weighted by ROI-specific attention onto the pixel-level image grid, yielding heatmaps that highlight where each ROI attends. Then, we overlaid the ROI attention maps on NSD images for representative category-selective regions in human brain.

## 6 RESULTS

### 6.1 DECODING PERFORMANCE

We evaluated our approach on 8 image quality metrics (Section 5.2), comparing it against prior single-subject decoding methods, including *Cortex2Image* (Gu et al., 2024), Takagi & Nishimoto (2023), *Brain Diffuser* (Ozcelik & VanRullen, 2023), *MindEye1* (Scotti et al., 2023), and *DREAM* (Xia et al., 2024). We further report results from recent multi-subject models, *MindFormer* (Han et al., 2024) and *MindBridge* (Wang et al., 2024a), which were trained using single-subject datasets for fair comparison. Also, we established a baseline model for each subject by retrieving an image from 1.3 million ImageNet images (Deng et al., 2009) whose predicted neural activity from our encoder best correlates with the ground truth fMRI response, following the spirit of earlier feature-matching based decoding approaches inspired by Kay et al. (2008). Examples of the baseline are shown in Appendix B.

From Fig. 5 (a), we observe that *NeuroAdapter* achieves competitive performance with, and in some cases surpasses, embedding-aligned approaches on high-level semantic metrics. This pattern suggests that despite its simplicity, our model is particularly effective at capturing semantic content encoded in the fMRI signals without the use of an intermediate representation (Fig. 4, Appendix A).

Additionally, our approach also captures low-level metrics reasonably well compared to the baseline retrieval method, although these improvements are more modest compared to those reported by other methods. To better understand it, we compared our performance with *Brain Diffuser* models using different embedding spaces. As evident in Fig. 5 (b), the better performance comes from the separate model pathway for predicting low-level latent features and removing them, as in the case of *Brain-Diffuser w/o VDVAE*, making their performance comparable to ours on low-level metrics. By design, we chose not to include such a pathway in *NeuroAdapter* and instead have a more direct and interpretable link between brain activity and image reconstruction (see Section 6.2).

We further compare how well brain activity predicted from the decoded images matches the measured fMRI responses (Fig. 3 (b) in red). We also report the correlation between the predicted activity for the ground truth image and the corresponding measured fMRI responses (Fig. 3 (b) in blue). This figure shows that the decoded images have visual properties sufficient to elicit predicted neural activity similar to the activity evoked by original image, further strengthening our decoding results.

We report performance of our model on two additional datasets, NSD-Imagery and Deeprecon, (Kneeland et al., 2025; Shen et al., 2019b) with quantitative and qualitative results reported in the Appendix E, F, Q, and R. On NSD-Imagery, *NeuroAdapter* demonstrates comparable generalization ability across both mental imagery and vision trials compared to existing work, especially for high-level semantic metrics. Experiments on Deeprecon, where training and test classes are disjoint, suggest that the model is able to infer not only category-level information but also finer low-level visual properties such as shape (e.g., coin), orientation (e.g., instrument), and color (e.g., reddish reconstructions for pink artificial shapes). To our knowledge, no existing diffusion-based decoding pipelines have been quantitatively evaluated on Deeprecon, and we provide our results as a baseline for future research.

### 6.2 DECODING INTERPRETABILITY

In this section, we visualize and analyze how brain representations influence the generative process with cross attention in *NeuroAdapter*. As mentioned in Section 4, our proposed *IBBI* framework

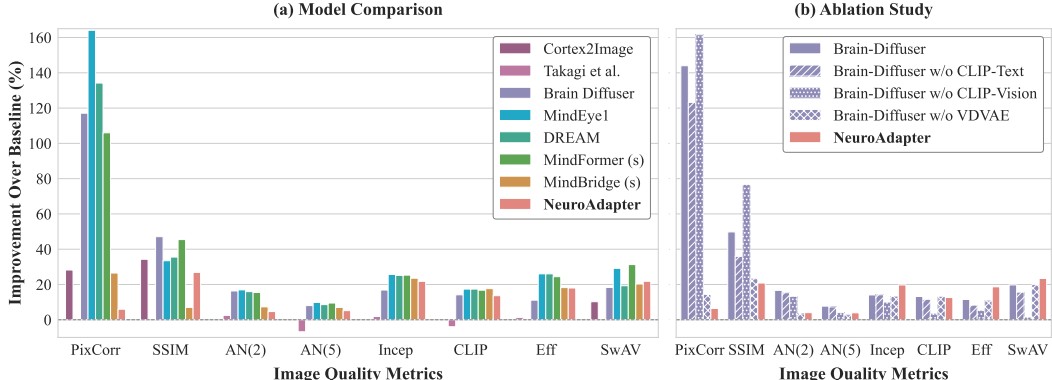

Figure 5: **Model Comparison.** Decoding performance across eight image quality metrics, comparing prior approaches and our method. To ensure fair comparison, results are shown as relative improvements over a subject-specific ImageNet-retrieval baseline. (a) *NeuroAdapter* achieves competitive performance with embedding-aligned approaches, particularly on high-level semantic metrics. (b) Comparison with *Brain Diffuser* variants shows that their advantage on low-level metrics arises from a dedicated pathway for predicting latent visual features (VDVAE), whereas removing this pathway yields performance on low-level metrics comparable to ours.

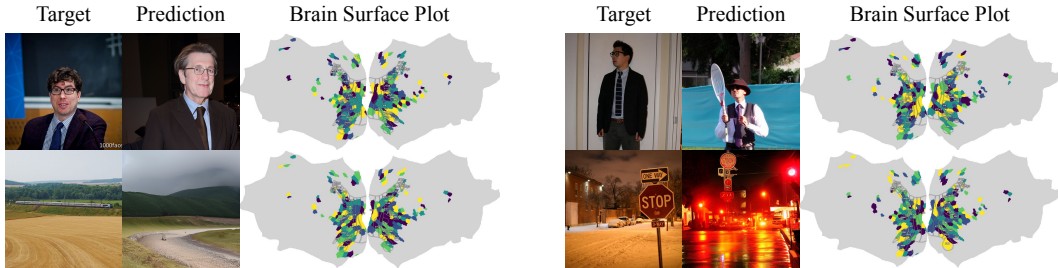

Figure 6: **Example projections of averaged parcel contribution vector onto the cortical surface across denoising steps.** Yellow colors denote parcels with strong influence across the denoising trajectory, while blue regions have a weaker contribution.

provides two complementary perspectives, showing how different brain regions contribute to visual reconstruction and where those ROIs direct their attention in the pixel-level stimulus space.

**Brain-directed View.** Based on the parcel contribution vector, we averaged $\mathrm{B}^{(t)}$ across timesteps to obtain a global view $\overline{\mathrm{B}^{(t)}}$ summarizing parcel contributions throughout the generative process. The 200 parcels and their corresponding contribution weights were projected onto the cortical surface for visualization. For easy interpretation through the visualization (Fig. 6), we ranked the parcels by their average contribution and divided them into five partitions (top 20%, 20–40%, 40–60%, 60–80%, and bottom 20%). This partitioning highlights the relative importance of different cortical regions, enabling us to identify high-impact parcels that dominate the generative trajectory and low-impact parcels that play only minor roles.

**Image-directed View.** Here, we visualize the ROI attention maps (RAM) across generative timesteps for representative category-selective regions, including *Face*, *Body*, *Scene*, and *Word*. Fig. 7 reveals how different cortical ROIs guide attention toward distinct spatial locations in the image during the unfolding denoising process, thereby linking regional neural signals to specific pixel-level features. Additional examples of ROI attention maps are provided in Appendix N.

To further evaluate *RAMs*, we computed Intersection-over-Union (IoU) and Dice scores between ROI-specific IBBI masks and semantic segmentation masks from Segment Anything 3 (SAM3; (Carion et al., 2025)), which serve as pseudo–ground truth. For *IBBI* masks, each ROI produces

a 2D attention map over denoising steps. We followed the approach from (Tang et al., 2023b) to obtain binary masks representing ROI-specific attended regions. A whole-image mask was used as an "attend everywhere" baseline. The quantitative results in Table 13 of Appendix O show that Face, Body, and Word ROIs have substantially higher IoU and Dice scores with *IBBI* masks compared to the whole-image baseline. Scene masks returned by SAM3 typically cover large, contiguous background regions, which inflates IoU/Dice for the whole-image baseline because most pixels belong to the "scene" class. Example segmentation maps are also included in Appendix O.

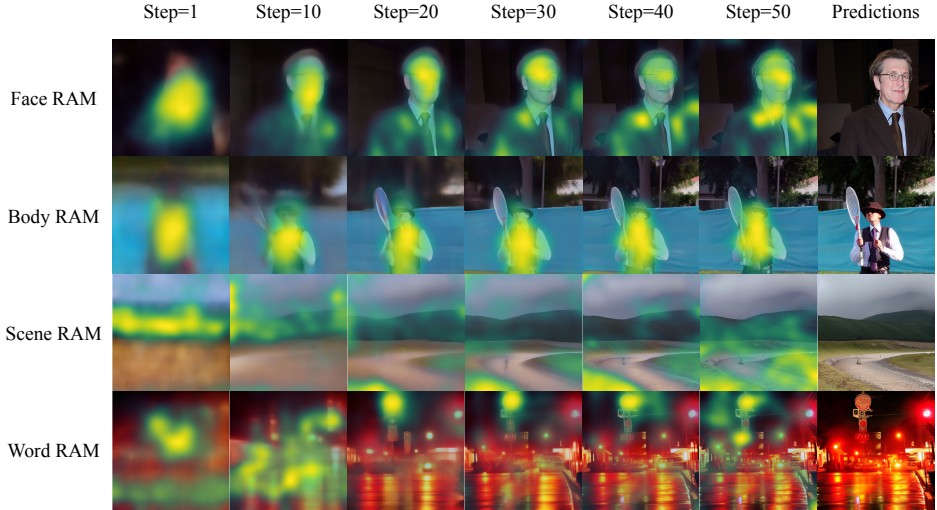

Figure 7: **ROI attention map dynamics across generative timesteps.** In early denoising steps, when the image is still highly blurred, maps are broadly distributed; as the denoising progresses and structure emerges, the attention becomes selective, converging on regions relevant to the content.

**Causal Perturbation Analysis.** Having the parcel-wise linear mapping further allows us to perform perturbation analysis, in which we masked specific ROIs and examined how this manipulation altered the reconstructed images. Consistent with the selectivity of different ROIs, we observe that masking low-level ROIs does not compromise the semantic content of the generated images, but masking high-level ROIs completely changes them (See Appendix P for ablation details).

## 7 Discussion

We present an effective end-to-end brain-decoding framework that directly conditions the diffusion denoising process on brain activity, bypassing intermediate feature spaces and enabling both effective decoding and mechanistic interpretability. Our results show that this approach achieves competitive reconstruction quality, particularly on high-level semantic metrics. Due to the stochastic nature of the diffusion model, we observe large variability in the quality of the generated images. While our encoder based selection addresses this limitation to some extent, future work will have to better understand the mapping from brain activity to images and make model performance more consistent. We believe this will be a great use case for interpretability methods in this domain.

Meanwhile, we notice that current brain-decoding benchmarks may be approaching saturation when evaluated solely through image quality metrics. Improvements in these scores do not necessarily reflect faithful brain decoding, as they may also result from stronger alignment with pretrained embedding spaces or simply the use of more powerful generative models. Therefore, our *IBBI* framework provides a complementary perspective, aiming to reveal how cortical parcels contribute to and shape the unfolding generative process, thereby linking brain activity and image features in a bi-directional manner. Looking ahead, future progress in brain decoding will depend on both methodological advances and richer interpretability frameworks, moving beyond metric-driven evaluation toward a deeper understanding of the neural–generative interface.

## ACKNOWLEDGMENTS

Research reported in this publication was supported in part by the National Institute of Neurological Disorders and Stroke of the National Institutes of Health under award numbers 1RF1NS128897 and 4R01NS128897. The content is solely the responsibility of the authors and does not necessarily represent the official views of the National Institutes of Health.

## REPRODUCIBILITY STATEMENT

We have made every effort to ensure the reproducibility of our work. Details of the datasets used, including NSD core, NSD imagery and Deeprecon, are provided in Section 5.1. The architecture of *NeuroAdapter*, training objectives and evaluation setup are described in Section 3. Our interpretability framework (*IBBI*) is fully specified in Section 4, including the mathematical definitions. We also provide results of ablation studies in appendices to verify the robustness of our results. For computational reproducibility, our models were trained on a university GPU cluster with 2 NVIDIA L40 GPUs. Each model was trained for 300 epochs with a batch size of 16, requiring approximately 25 hours of training time. Source code, along with instructions for reproducing all experiments, is available at `https://github.com/kriegeskorte-lab/NeuroAdapter`.

## THE USE OF LARGE LANGUAGE MODELS (LLMS)

Large Language Models (LLMs) were used in this project as general-purpose assistant tools. Specifically, we used GitHub Copilot with Claude 3.7 to help sort and refactor code for readability and debugging during the research process, and used OpenAI ChatGPT-5 to polish the writing for clarity and effective communication of our ideas. No part of the model design, experimental results, or scientific conclusions depended on LLMs.

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

APPENDIX

## A EXAMPLES OF DECODED STIMULI ON NSD

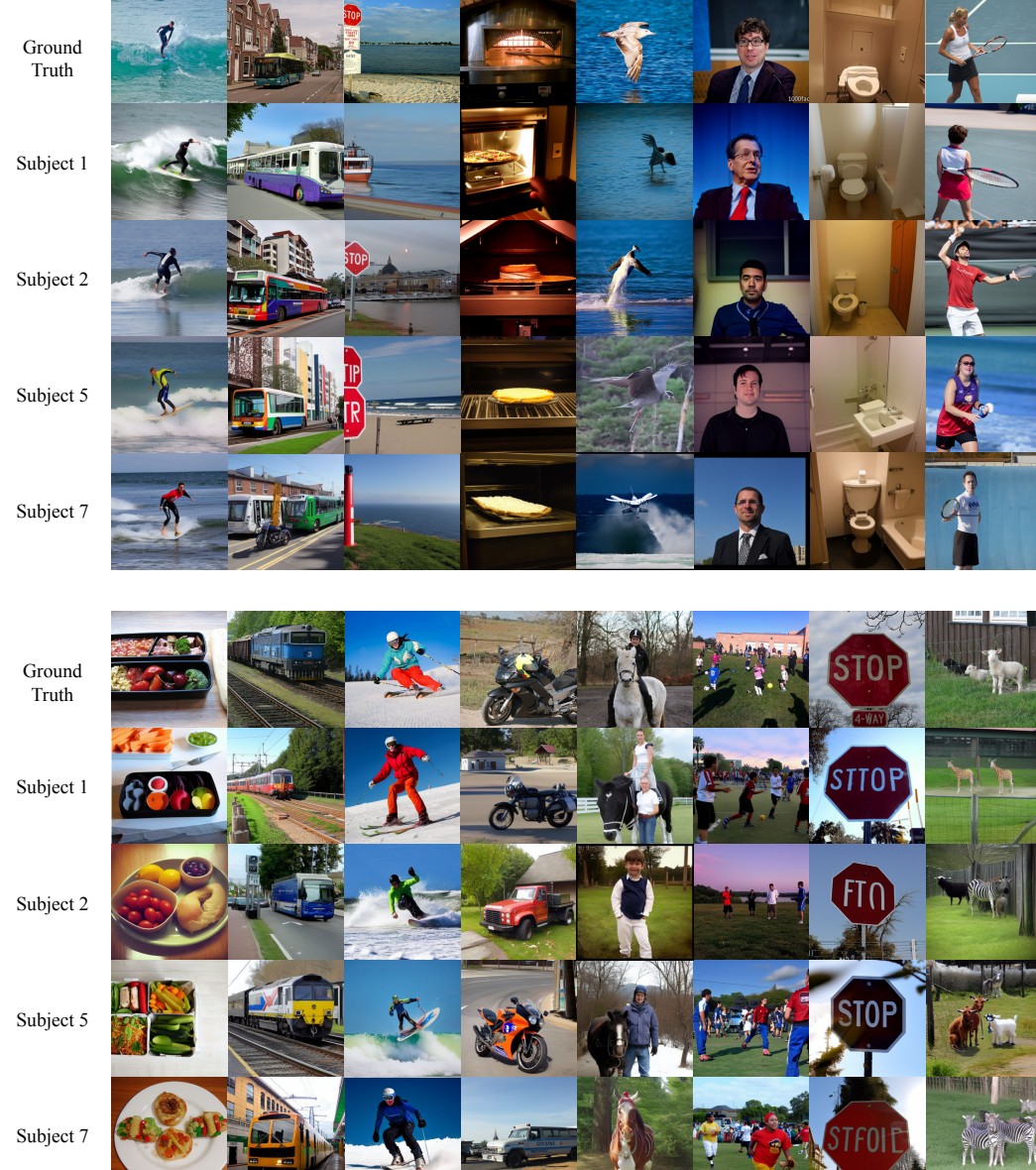

Figure 8: Examples of ground truth with corresponding decoded stimuli across subjects

## B    EXAMPLES OF IMAGENET RETRIEVAL BASELINES

We present shared retrieved images across 4 subjects in this figure. In our experiment, we created and evaluated baselines separately for each subject.

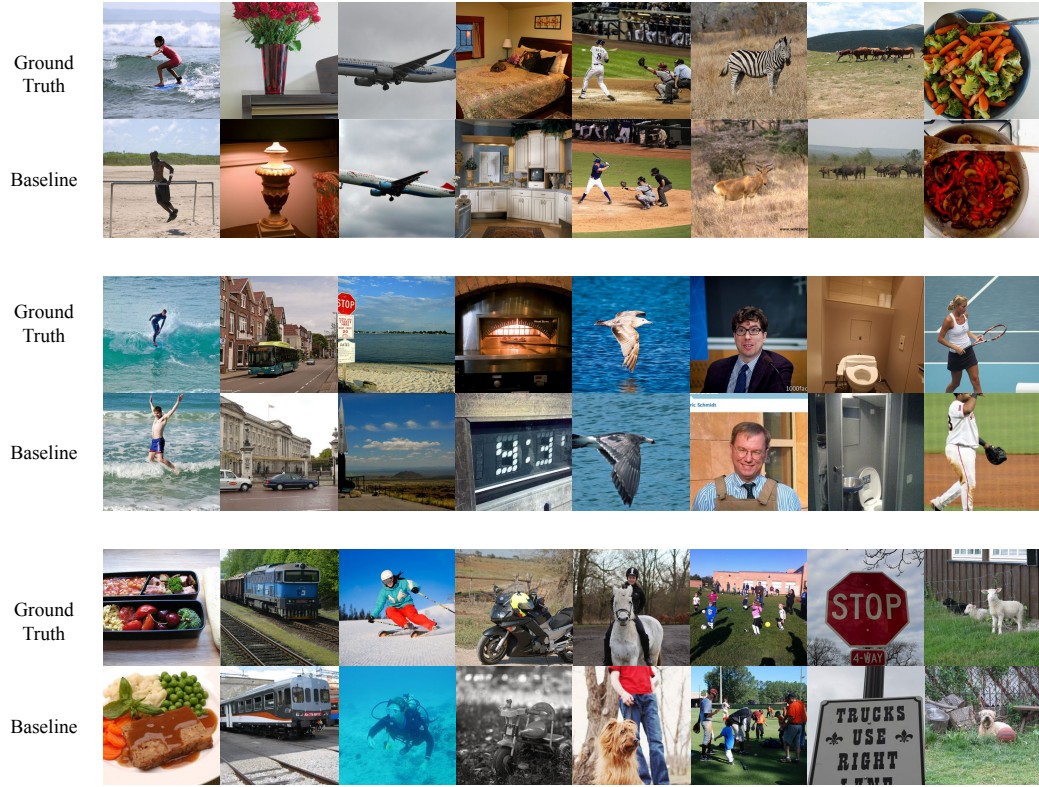

Figure 9: Ground Truth vs. ImageNet Retrieval Baselines.

## C    SCHAEFER PARCELLATION

To represent brain activity at the regional level, we adopt the Schaefer cortical parcellation (Fig. 10). This provides a functional subdivision of the cortex derived from large-scale resting-state fMRI. In our experiments, we compute vertex-wise Signal-to-Noise Ratio (SNR) and select top 100 parcels per hemisphere with the highest average SNR.

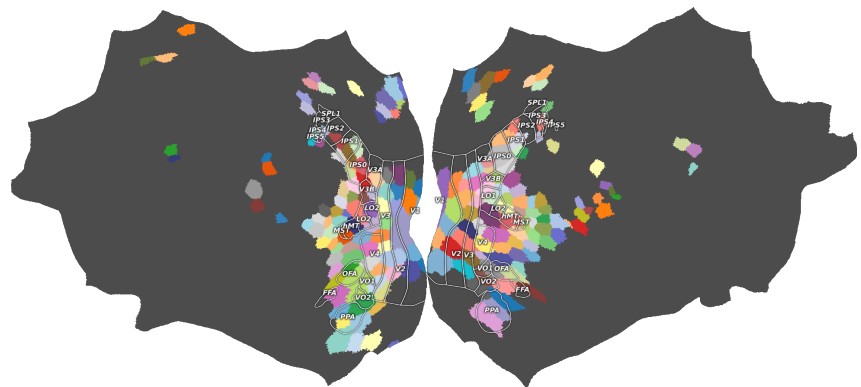

Figure 10: Top-100-SNR Parcels for each brain hemisphere displayed on the cortical surface.

# D MODEL PERFORMANCE ON NSD

Table 1: **Performance across different image quality metrics**

| Method | Low-Level | | | | High-Level | | | |
|---|---|---|---|---|---|---|---|---|
| | PixCorr ↑ | SSIM ↑ | Alex(2) ↑ | Alex(5) ↑ | Incep ↑ | CLIP ↑ | Eff ↓ | SwAV ↓ |
| Baseline | .117 | .242 | 80.98% | 89.05% | 74.62% | 80.16% | .872 | .518 |
| Cortex2Image | .150 | .325 | – | – | – | – | .862 | .465 |
| Takagi et al. | – | – | 83.0% | 83.0% | 76.0% | 77.0% | – | – |
| Brain Diffuser | .254 | .356 | 94.2% | 96.2% | 87.2% | 91.5% | .775 | .423 |
| MindEye1 | .309 | .323 | 94.7% | 97.8% | 93.8% | 94.1% | .645 | .367 |
| DREAM | .274 | .328 | 93.9% | 96.7% | 93.4% | 94.1% | .645 | .418 |
| MindFormer (s) | .241 | .352 | 93.5% | 97.5% | 93.5% | 93.6% | .659 | .356 |
| MindBridge (s) | .148 | .259 | 86.9% | 95.3% | 92.2% | 94.3% | .713 | .413 |
| NeuroAdapter | .124 | .307 | 84.54% | 93.48% | 90.79% | 90.97% | .716 | .408 |

Table 2: **NeuroAdapter vs. Brain-Diffuser performance on data from Subject 1**

| Method | Low-Level | | | | High-Level | | | |
|---|---|---|---|---|---|---|---|---|
| | PixCorr ↑ | SSIM ↑ | Alex(2) ↑ | Alex(5) ↑ | Incep ↑ | CLIP ↑ | Eff ↓ | SwAV ↓ |
| Baseline | .125 | .245 | 82.88% | 90.41% | 76.98% | 81.72% | .868 | .517 |
| BD | .305 | .367 | 96.7% | 97.4% | 87.8% | 92.5% | .768 | .415 |
| BD w/o CLIP-Text | .279 | .333 | 95.6% | 97.0% | 87.9% | 91.2% | .796 | .436 |
| BD w/o CLIP-Vision | .327 | .433 | 93.9% | 94.1% | 84.7% | 84.5% | .821 | .509 |
| BD w/o VDVAE | .143 | .302 | 85.6% | 93.0% | 87.3% | 92.6% | .775 | .414 |
| NeuroAdapter | .133 | .296 | 86.22% | 93.96% | 92.15% | 92.03% | .706 | .396 |

# E MODEL PERFORMANCE ON NSD-IMAGERY

Table 3: **NSD-Imagery: Mental Imagery vs. Vision Trials**

| Method | Low-Level | | | | High-Level | | | |
|---|---|---|---|---|---|---|---|---|
| | PixCorr ↑ | SSIM ↑ | Alex(2) ↑ | Alex(5) ↑ | Incep ↑ | CLIP ↑ | Eff ↓ | SwAV ↓ |
| **NSD-Imagery Mental Imagery Trials** | | | | | | | | |
| MindEye1 | .086 | .349 | **59.56%** | **61.00%** | 52.03% | 54.72% | .948 | .564 |
| Brain Diffuser | .064 | .401 | 52.14% | 58.35% | 52.73% | 54.07% | **.935** | .585 |
| iCNN | **.108** | .340 | 50.57% | 55.25% | 49.39% | 41.72% | .994 | **.560** |
| MindEye2 | .036 | .414 | 47.60% | 55.38% | 46.02% | 50.78% | .966 | .591 |
| Takagi et al. | -.006 | **.455** | 41.88% | 40.19% | 43.26% | 40.08% | .976 | .606 |
| NeuroAdapter | .037 | .312 | 58.90% | 58.71% | **57.26%** | **60.04%** | .970 | .603 |
| **NSD-Imagery Vision Trials** | | | | | | | | |
| MindEye1 | .218 | .412 | 73.56% | 80.81% | 62.44% | 65.34% | **.881** | **.510** |
| Brain Diffuser | .107 | .455 | 60.34% | 72.84% | 60.95% | 58.31% | .908 | .555 |
| iCNN | **.224** | .385 | 71.67% | **81.35%** | 61.16% | 49.03% | .926 | .524 |
| MindEye2 | .161 | **.480** | 70.10% | 77.52% | 62.69% | 65.93% | .886 | .512 |
| Takagi et al. | -.013 | .412 | 41.55% | 39.26% | 39.26% | 43.01% | .969 | .610 |
| NeuroAdapter | .077 | .342 | **75.76%** | 78.54% | **68.18** | **70.45%** | .945 | .576 |

## F MODEL PERFORMANCE ON DEEPRECON

Table 4: **Performance on Deeprecon natural images**

| Condition:
Token dropout, num of preds | Low-Level | | | | High-Level | | | |
|---|---|---|---|---|---|---|---|---|
| | PixCorr↑ | SSIM↑ | Alex(2)↑ | Alex(5)↑ | Incep↑ | CLIP↑ | Eff↓ | SwAV↓ |
| wo/ keep low-level, 4 | .087 | .309 | 77.7% | 86.6% | 74.2% | 81.0% | .902 | .552 |
| wo/ keep low-level, 8 | .093 | .310 | 79.1% | 87.6% | **74.8%** | 81.3% | .898 | **.545** |
| wo/ keep low-level, 16 | **.102** | .314 | **80.0%** | **88.8%** | 74.6% | **81.7%** | **.892** | .546 |
| keep low-level, 4 | .088 | **.316** | 78.8% | 86.9% | 73.4% | 80.8% | .908 | .552 |
| keep low-level, 8 | .084 | .314 | 79.5% | 87.0% | 72.6% | 81.1% | .907 | .550 |
| keep low-level, 16 | .081 | .311 | 79.9% | 87.0% | 71.2% | 80.0% | .908 | .553 |

Table 5: **Performance on Deeprecon artificial shapes**

| Conditions
Token dropout, num of preds | Low-Level | | | | High-Level | | | |
|---|---|---|---|---|---|---|---|---|
| | PixCorr↑ | SSIM↑ | Alex(2)↑ | Alex(5)↑ | Incep↑ | CLIP↑ | Eff↓ | SwAV↓ |
| wo/ keep low-level, 4 | .050 | .484 | 63.0% | 55.7% | 51.2% | 52.1% | .960 | .622 |
| wo/ keep low-level, 8 | .062 | **.485** | **63.2%** | **56.4%** | **53.2%** | 50.6% | .958 | .622 |
| wo/ keep low-level, 16 | **.067** | .477 | 57.1% | 53.4% | 53.1% | 52.4% | .961 | .626 |
| keep low-level, 4 | .057 | .470 | 59.7% | 55.0% | 51.3% | 52.3% | .958 | .622 |
| keep low-level, 8 | .056 | .475 | 62.6% | 54.7% | 51.2% | **52.8%** | .958 | **.621** |
| keep low-level, 16 | .057 | .478 | 61.8% | 55.8% | 52.9% | 51.8% | **.955** | .622 |

## G ABLATION STUDY: BRAIN TOKEN DROPOUT

We conducted an ablation study to evaluate the effect of the proposed fMRI token dropout (TD) strategy in training on decoding performance. As shown in Table 6, removing token dropout substantially compromised performance across almost all metrics.

Table 6: **Effect of parcel-wise token dropout (TD) on model performance**

| Conditions | Low-Level | | | | High-Level | | | |
|---|---|---|---|---|---|---|---|---|
| | PixCorr↑ | SSIM↑ | Alex(2)↑ | Alex(5)↑ | Incep↑ | CLIP↑ | Eff↓ | SwAV↓ |
| without TD | .038 | **.307** | 67.4% | 75.5% | 61.2% | 64.8% | .974 | .666 |
| with TD | **.133** | .296 | **86.22%** | **93.96%** | **92.15%** | **92.03%** | **.706** | **.396** |

## H ABLATION STUDY: NUMBER OF CONDITION DIMENSION

Table 7: **Effect of different condition dimension (CD) on model performance**

| Conditions | Low-Level | | | | High-Level | | | |
|---|---|---|---|---|---|---|---|---|
| | PixCorr↑ | SSIM↑ | Alex(2)↑ | Alex(5)↑ | Incep↑ | CLIP↑ | Eff↓ | SwAV↓ |
| CD = 1024 | .116 | **.303** | 81.49% | 91.64% | 88.17% | 90.33% | .742 | .432 |
| CD = 960 | .134 | .302 | 85.49% | 93.38% | 91.32% | 90.62% | .720 | .410 |
| CD = 768 | .133 | .296 | 86.22% | 93.96% | **92.15%** | **92.03%** | **.706** | **.396** |
| CD = 576 | .132 | .297 | **86.31%** | 93.87% | 90.42% | 90.76% | .712 | .400 |
| CD = 384 | **.136** | .301 | 85.65% | 94.16% | 89.78% | 90.13% | .725 | .411 |
| CD = 192 | .122 | .290 | 85.25% | **94.17%** | 91.33% | 90.78% | .718 | .417 |

# I   ABLATION STUDY: NUMBER OF HIGHEST-SNR PARCELS

Table 8: **Effect of number of highest-SNR parcels ($p$) on model performance**

| Conditions | Low-Level | | | | High-Level | | | |
|---|---|---|---|---|---|---|---|---|
| | PixCorr ↑ | SSIM ↑ | Alex(2) ↑ | Alex(5) ↑ | Incep ↑ | CLIP ↑ | Eff ↓ | SwAV ↓ |
| $p = 100$ | .106 | .296 | 84.87% | 93.65% | 88.52% | 90.41% | .730 | .414 |
| $p = 200$ | **.133** | .296 | **86.22%** | **93.96%** | **92.15%** | **92.03%** | **.706** | **.396** |
| $p = 500$ | .094 | **.307** | 77.83% | 88.93% | 85.84% | 88.66% | .767 | .449 |
| $p = 1000$ | .086 | .297 | 76.37% | 88.09% | 83.95% | 84.89% | .778 | .457 |

# J   ABLATION STUDY: PARCEL-WISE LINEAR MAPPER

Table 9: **Effect of parcel-wise linear mapper (LM) on model performance**

| Conditions | Low-Level | | | | High-Level | | | |
|---|---|---|---|---|---|---|---|---|
| | PixCorr ↑ | SSIM ↑ | Alex(2) ↑ | Alex(5) ↑ | Incep ↑ | CLIP ↑ | Eff ↓ | SwAV ↓ |
| with LM | **.133** | .296 | **86.22%** | **93.96%** | **92.15%** | **92.03%** | **.706** | **.396** |
| w/o LM | .073 | **.318** | 73.81% | 82.47% | 76.20% | 78.11% | .843 | .515 |

# K   ABLATION STUDY: BRAIN ENCODER AS A RANKING TOOL

We further evaluate the role of the brain encoder as a selection mechanism for decoded stimuli. Table 10 shows that increasing the number of candidate predictions consistently improves decoding performance. In addition, we conduct an additional experiment in which each test sample is decoded eight times with different random initializations. We report image-quality metrics in Table 11 for three conditions: (i) the Highest-Corr candidate selected by the brain encoder, (ii) the Lowest-Corr candidate, and (iii) a Random candidate drawn uniformly from the eight samples. The brain encoder consistently improves performance relative to Lowest-Corr and Random selections, but Random images occasionally score higher on certain perceptual metrics, indicating that the encoder is not optimizing for image quality. Instead, it selects candidates that are most aligned with the neural data, highlighting its role as a neural-fidelity criterion rather than a perceptual metric booster.

Table 10: **Effect of encoder-based selection across different number of predictions**

| Conditions | Low-Level | | | | High-Level | | | |
|---|---|---|---|---|---|---|---|---|
| | PixCorr ↑ | SSIM ↑ | Alex(2) ↑ | Alex(5) ↑ | Incep ↑ | CLIP ↑ | Eff ↓ | SwAV ↓ |
| num of preds = 1 | .104 | .292 | 79.0% | 90.1% | 89.5% | 90.8% | .729 | .417 |
| num of preds = 2 | .105 | .292 | 82.6% | 91.7% | 89.8% | 88.7% | .733 | .416 |
| num of preds = 4 | .120 | .293 | 84.0% | 93.5% | 90.1% | 91.5% | .725 | .408 |
| num of preds = 8 | **.133** | **.296** | **86.22%** | **93.96%** | **92.15%** | **92.03%** | **0.706** | **.396** |

Table 11: **Results of encoder-based selection on 8 predictions per test sample.**

| Conditions | Low-Level | | | | High-Level | | | |
|---|---|---|---|---|---|---|---|---|
| | PixCorr ↑ | SSIM ↑ | Alex(2) ↑ | Alex(5) ↑ | Incep ↑ | CLIP ↑ | Eff ↓ | SwAV ↓ |
| Lowest Corr | 0.109 | **0.286** | 80.88% | 89.51% | 89.81% | 91.31% | 0.720 | 0.404 |
| Random | 0.130 | **0.286** | 84.00% | 91.30% | **92.05%** | 92.33% | 0.708 | 0.395 |
| Highest Corr | **0.139** | 0.285 | **88.02%** | **93.87%** | 91.91% | **93.01%** | **0.702** | **0.387** |

## L    ABLATION STUDY: NUMBER OF DENOISING STEPS IN REVERSED DIFFUSION PROCESS

Regarding the number of inference steps, we follow the default setting of 50 denoising steps used in Stable Diffusion for inference. Further, we evaluated decoding quality across a range of denoising steps (20–80). As shown in Table 12, performance remains highly stable around 50 steps, and no monotonic improvement is observed with more steps. These results indicate that the number of diffusion steps is not a sensitive hyperparameter in our pipeline, consistent with prior observations in diffusion-based brain decoding.

Table 12: **Decoding performance across different numbers of denoising steps.**

| Steps | PixCorr ↑ | SSIM ↑ | AN(2) ↑ | AN(5) ↑ | Incep ↑ | CLIP ↑ | Eff ↓ | SwAV ↓ |
|-------|-----------|--------|---------|---------|---------|--------|-------|--------|
| 20 | 0.137 | 0.291 | 87.92% | 95.46% | 90.49% | 90.34% | 0.707 | 0.396 |
| 30 | 0.135 | 0.291 | 87.93% | 94.50% | 90.90% | 90.93% | 0.706 | 0.391 |
| 40 | 0.142 | **0.299** | **88.06%** | **95.96%** | 91.58% | 91.83% | 0.705 | 0.388 |
| 50 | **0.143** | 0.289 | 87.96% | 95.42% | **91.72%** | 91.19% | 0.705 | 0.389 |
| 60 | 0.141 | 0.288 | 87.26% | 94.61% | 91.50% | **91.62%** | 0.702 | **0.386** |
| 70 | 0.137 | 0.284 | 87.40% | 94.85% | 91.46% | 91.10% | **0.700** | 0.389 |
| 80 | 0.138 | 0.285 | 87.13% | 94.47% | 90.77% | 90.43% | 0.704 | 0.389 |

## M    EXPLANATIONS OF MIN-SNR LOSS WEIGHTING

At each diffusion timestep $t$, the effective signal-to-noise ratio is defined as

$$\text{SNR}_t = \frac{\bar{\alpha}_t}{1 - \bar{\alpha}_t},$$

where $\bar{\alpha}_t$ denotes the cumulative product of noise scheduling coefficients.

Without reweighting, high-SNR steps (early timesteps) tend to dominate the mean squared error (MSE) loss, while low-SNR steps (late timesteps) provide weaker gradients despite being more challenging and important for generation.

Ideally, the model should learn more from low-SNR noisy samples rather than overfitting to the easier, cleaner ones. Min-SNR weighting balances this trade-off by rescaling the per-timestep loss with

$$w_t = \frac{\min(\text{SNR}_t, \gamma)}{\text{SNR}_t},$$

where $\gamma$ is a threshold hyperparameter (we set it to 5.0 in training).

# N ROI ATTENTION MAP VISUALIZATION

To better interpret the ROI attention maps, we connect them to well-established functional regions. Because the Schaefer parcellation does not provide anatomical or functional labels for individual parcels, we assigned labels by mapping parcels to the labels available in NSD. A parcel was assigned to a given label if more than 50% of its vertices overlapped with that region. Using this mapping, we visualize the attention maps of the corresponding ROIs on generated images, tracking how their spatial influence evolves from noisy to clean across timesteps.

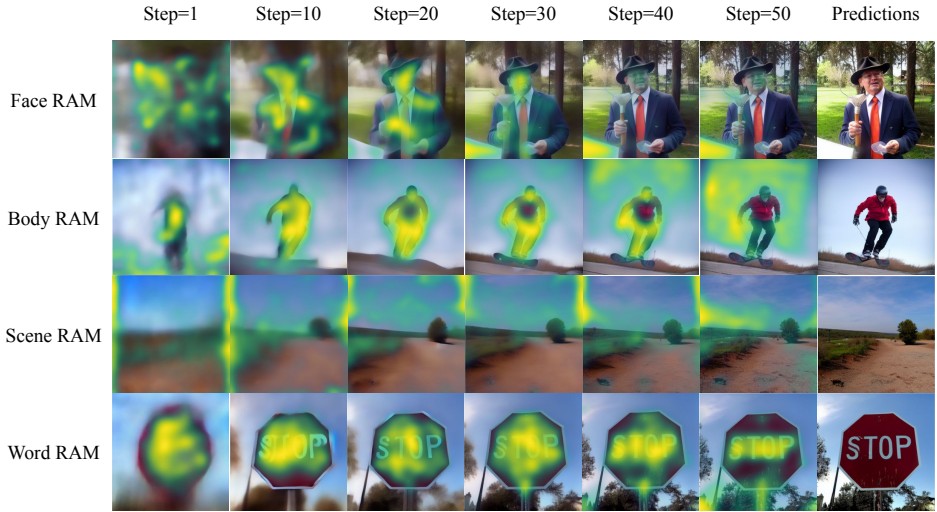

Figure 11: ROI attention map dynamics across generative timesteps

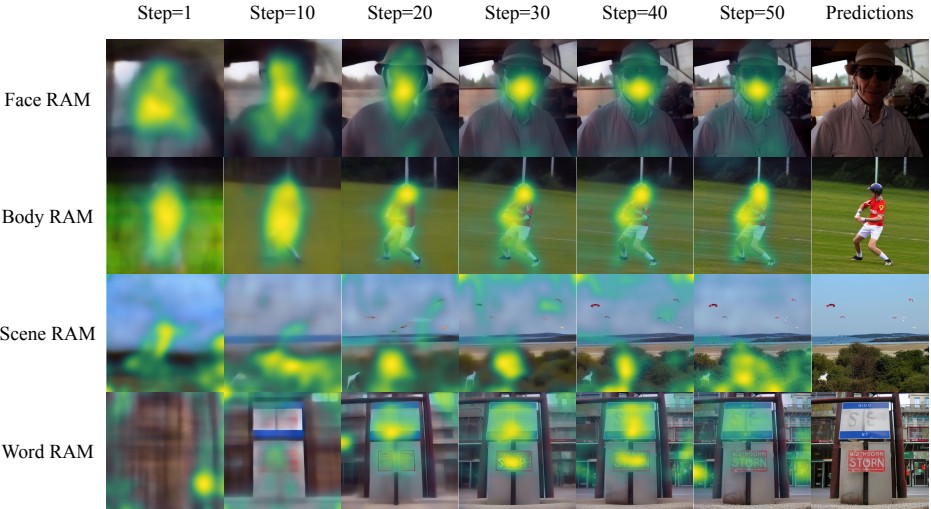

Figure 12: ROI attention map dynamics across generative timesteps

## O    EVALUATION OF *IBBI* ATTENTION MAPS

To quantify interpretability, we evaluate ROI-specific *IBBI* attention maps using Intersection-over-Union (IoU) and Dice scores, which measure the spatial overlap between predicted attention regions and semantic segmentation masks from the latest Segment Anything 3 (SAM3, (Carion et al., 2025)). SAM3 provides high-quality region segmentation and serves as pseudo–ground-truth for our generated images. Among the 515 test reconstructions, 236 images contain valid semantic regions, including 38 Face, 195 Body, 27 Scene, and 7 Word images.

For *IBBI* masks, each ROI produces a 2D attention map over denoising steps. We average attention across steps, normalize, and apply a 50% threshold to obtain binary masks representing ROI-specific attended regions, following the DAAM procedure (Tang et al., 2023b). As a baseline, we use a whole-image mask, representing the trivial strategy of "attending everywhere."

We compute Intersection-over-Union (IoU) and Dice between predicted masks and SAM3 masks. Face, Body, and Word ROIs show substantially higher IoU and Dice scores with *IBBI* attention maps compared to the whole-image baseline, demonstrating that *IBBI* reliably localizes semantically meaningful regions. Scene masks returned by SAM3 typically cover large, contiguous background regions, which inflates IoU/Dice for the whole-image baseline because most pixels belong to the "scene" class. In contrast, *IBBI* allocates attention selectively to diagnostic subregions rather than spreading uniformly across the entire background.

Table 13: **Evaluations on *IBBI* attention masks and baseline compared to SAM3 segmentations.**

| Method | IoU | | | | | Dice | | | |
|---|---|---|---|---|---|---|---|---|---|
| | Face | Body | Scene | Word | | Face | Body | Scene | Word |
| Whole-image Mask | 0.124 | 0.171 | **0.605** | 0.199 | | 0.213 | 0.276 | **0.717** | 0.313 |
| IBBI Attn Mask | **0.322** | **0.362** | 0.198 | **0.453** | | **0.459** | **0.504** | 0.314 | **0.595** |

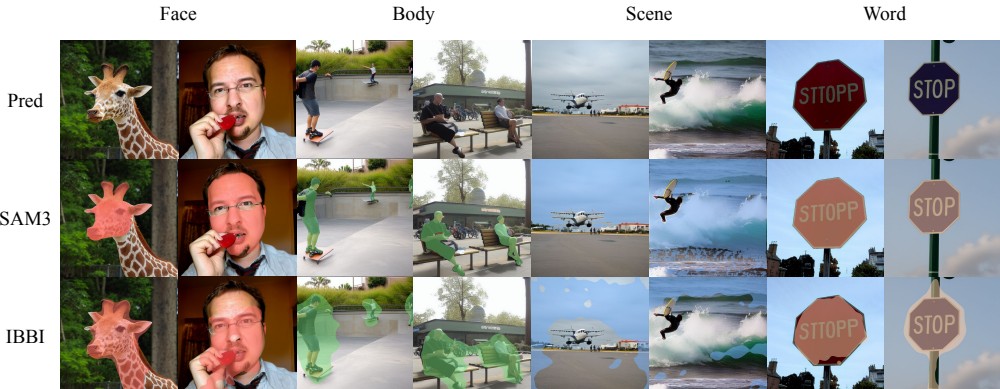

Figure 13: **Quantitative evaluation of ROI attention maps using SAM3 segmentation.**

## P  CAUSAL PERTURBATION WITH BRAIN ROI MASKING

We present causal perturbation by masking out the related parcels of ROIs. Because the Schaefer parcellation does not provide anatomical or functional labels for individual parcels, we assigned labels by mapping parcels to the labels available in NSD. A parcel was assigned to a given label if more than 50% of its vertices overlapped with that region. In 200 parcels with high SNR, 103 parcels were labeled for subject 1. Among them, 50 parcels were labeled as low-level ROIs, including V1, V2, V3, and V4, while 53 parcels were annotated as Face, Body, Scene and Word ROIs.

Table 14: **Effect of ROI masking (high-level vs. low-level) on model performance**

| Conditions | Low-Level | | | | High-Level | | | |
|---|---|---|---|---|---|---|---|---|
| | PixCorr ↑ | SSIM ↑ | Alex(2) ↑ | Alex(5) ↑ | Incep ↑ | CLIP ↑ | Eff ↓ | SwAV ↓ |
| No Masking | **.133** | **.296** | **86.22%** | **93.96%** | **92.15%** | **92.03%** | **.706** | **.396** |
| LL ROI Masking | .119 | .290 | 66.13% | 78.38% | 73.15% | 74.20% | .891 | .535 |
| HL ROI Masking | .019 | .289 | 55.42% | 58.34% | 50.70% | 50.37% | .981 | .641 |

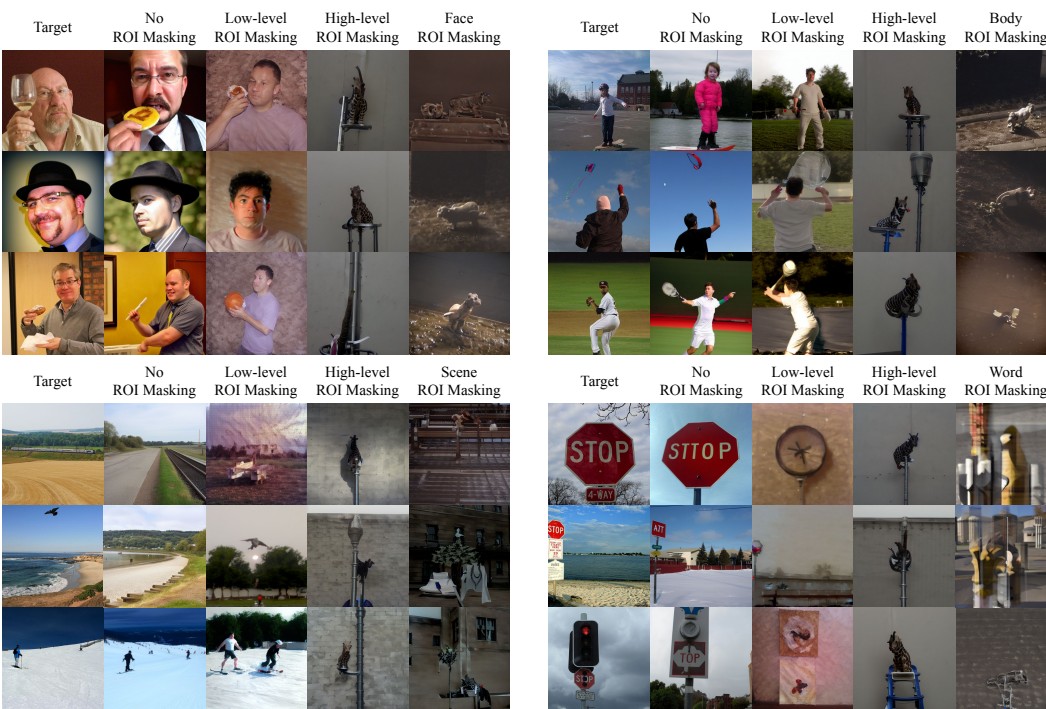

Figure 14: Visual Effect of brain masking on different ROIs

## Q    EXAMPLES OF DECODED STIMULI ON NSD IMAGERY DATASET

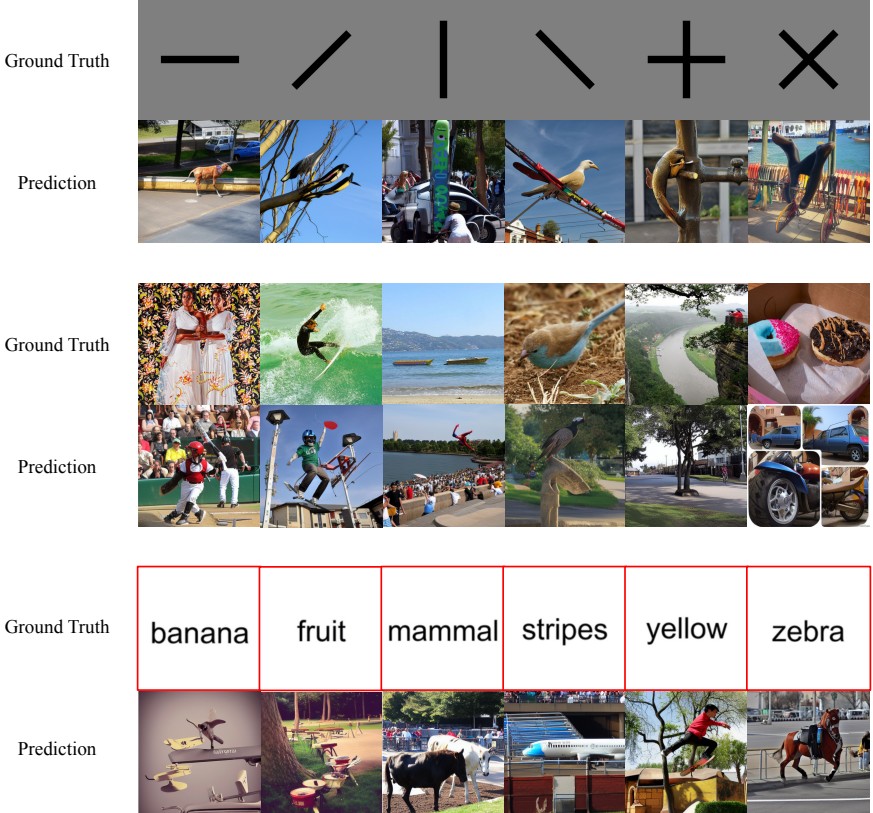

Figure 15: Best decoded examples on NSD-Imagery mental imagery task

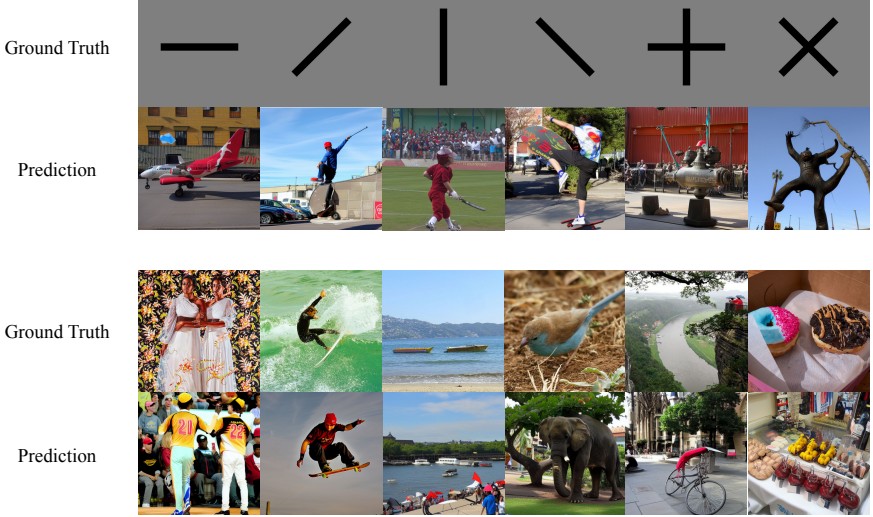

Figure 16: Best decoded examples on NSD-Imagery vision task

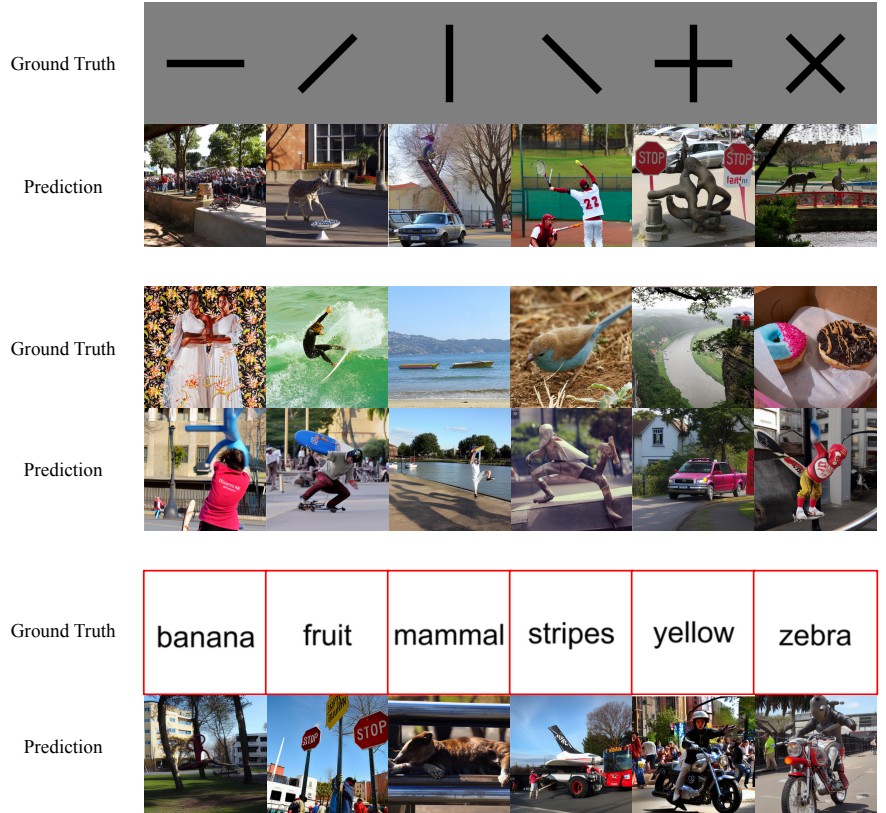

Figure 17: Worst decoded examples on NSD-Imagery mental imagery task

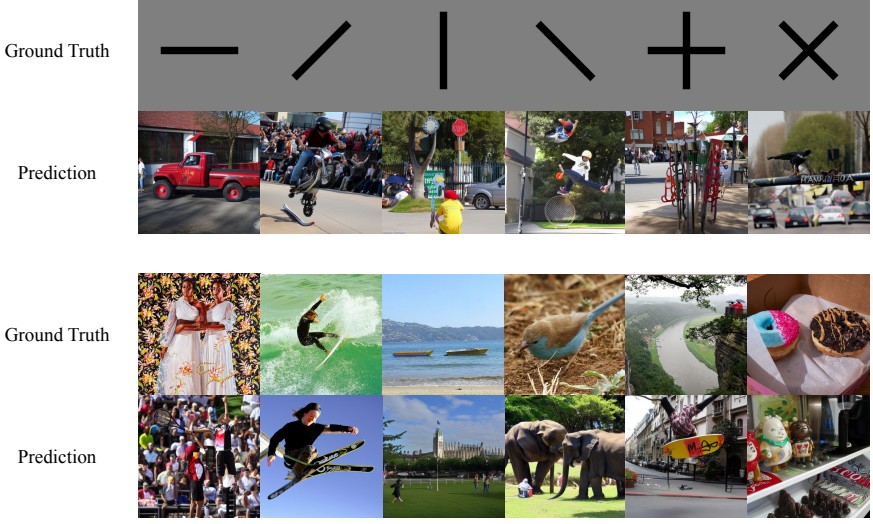

Figure 18: Worst decoded examples on NSD-Imagery vision task

## R    EXAMPLES OF DECODED STIMULI ON DEEPRECON DATASET

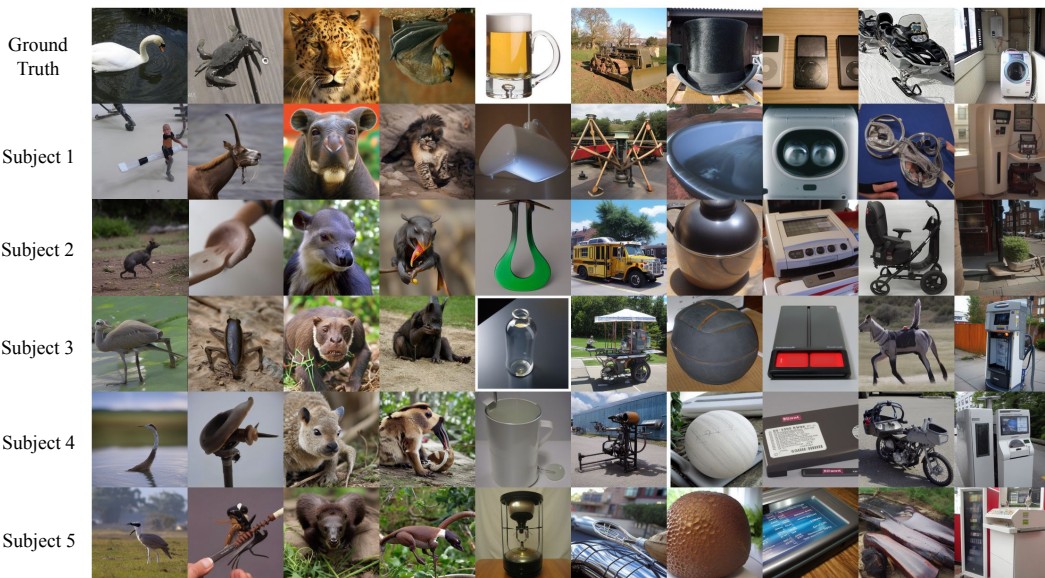

Figure 19: Decoded examples on Deeprecon natural image dataset

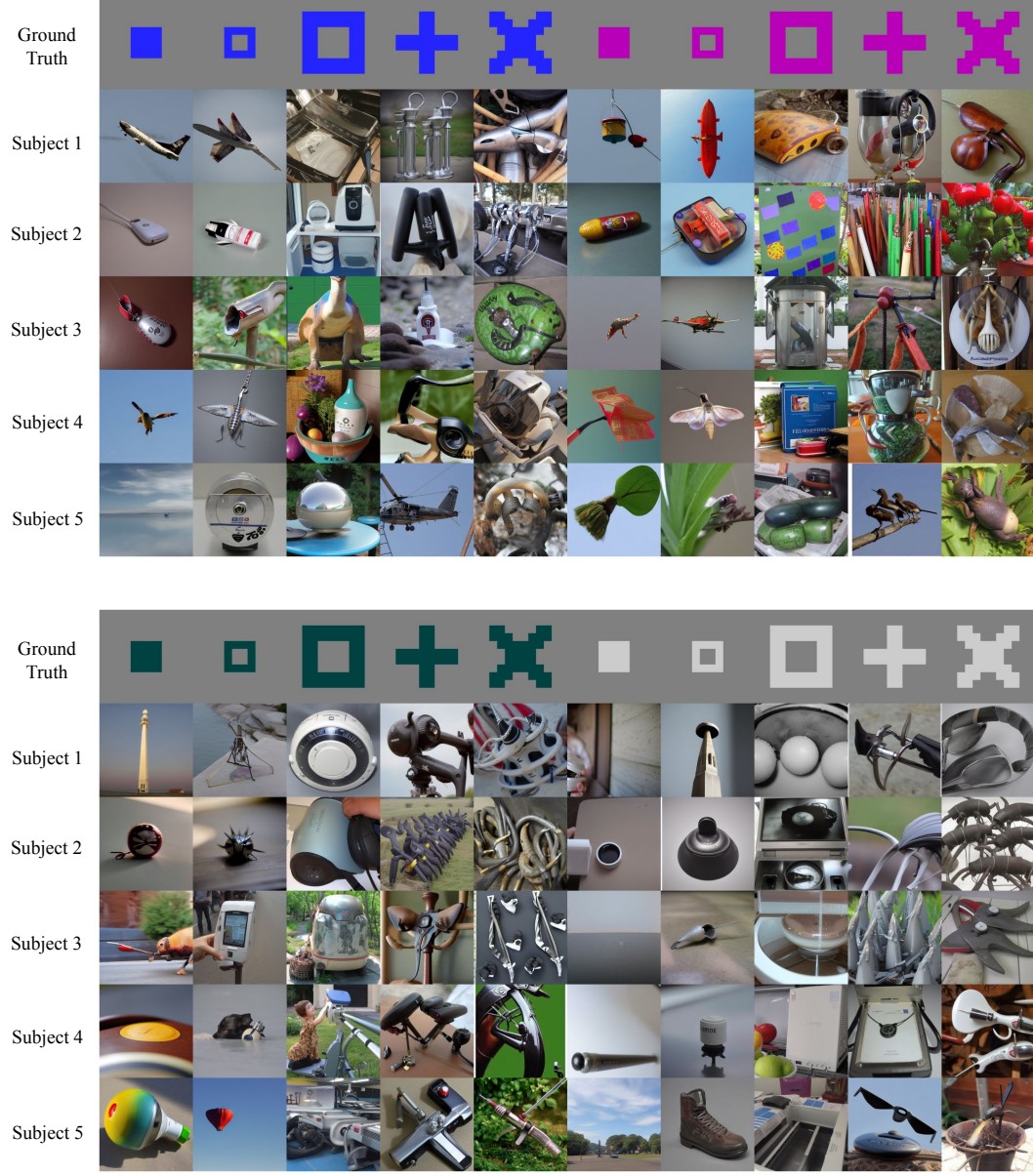

Figure 20: Decoded examples on Deeprecon artificial shape image dataset

