# OpenReview forum: "Towards Interpretable Visual Decoding with Attention to Brain Representations"
_ICLR.cc/2026/Conference — ICLR 2026 Poster_

### Official Review · Reviewer_5Pr2 · 2025-10-15

**Soundness:** 3
**Presentation:** 4
**Contribution:** 2
**Rating:** 6
**Confidence:** 5

**Summary:**

This paper “Towards Interpretable Visual Decoding with Attention to Brain Representations” introduces NeuroAdapter, an end-to-end fMRI-to-image decoding framework that directly conditions a latent diffusion model on brain-derived tokens, bypassing intermediate image or text embedding spaces like CLIP. The approach integrates fMRI parcel-wise embeddings into Stable Diffusion via IP-Adapter–style cross-attention, allowing the model to learn mappings from cortical representations to visual stimuli. To interpret the generative process, the authors introduce the IBBI ramework, which analyzes cross-attention dynamics during denoising to reveal how specific cortical regions influence image formation. NeuroAdapter achieves nice results on NSD, NSD-Imagery, and DeepRecon datasets, and provides meaningful visualizations linking cortical regions to specific visual features. I think the work is a solid piece but lacks some comparison with recent work that could shadow his novelty.

**Strengths:**

Quite elegant approach! The model removes the intermediate embedding bottleneck and directly links fMRI data to the diffusion process, addressing a key interpretability limitation in current brain-to-image pipelines.

The IBBI framework offers a structured and quantitative way to visualize brain–image interactions via cross-attention.

NeuroAdapter matches or exceeds existing diffusion-based decoders on several high-level metrics, even without CLIP conditioning. However it's not the best-in class (author don't claim that). I don't think this is really important given the technical contribution, interpretability gains and the fact the metrics are kind of satured.

The method performs reasonably well on imagery and DeepRecon datasets, suggesting decent cross-distribution generalization.

Overall, the paper is well-organized, includes ablations, clear diagrams, and code availability.

**Weaknesses:**

While the paper is solid and promising, several important references and comparisons are missing or insufficiently discussed:
1) Early encoder-based visual decoding: Nishimoto et al. (2011) pioneered similar end-to-end fMRI-to-video approaches, and this line of work should be acknowledged to contextualize the contribution, since Encoder here is used in a new fashion as a rescorer.

2)Language decoding parallels: Huth and Antonello’s works on encoder-based and interpretable language decoding provide useful methodological analogies that could enrich the interpretability discussion.

3)Interpretability precedents: Antonello et al. introduced early interpretability frameworks for language-to-brain decoding that align conceptually with IBBI’s aims.

4) Cross-subject generalization: Ferrante et al.’s work on NSD cross-subject decoding is relevant for comparison and could be informative for assessing NeuroAdapter’s generalization strategy.

5) Recent concurrent diffusion-based methods: The paper does not compare against or even mention Dynadiff (Meta, 2025), which jointly fine-tunes diffusion models with a brain module (akin to the IP-adapter here), achieving similar or superior results.

6) Interpretability-focused diffusion pipelines: BrainDIVE and BrainScuba combine diffusion models and neural encoders for interpretability analysis, conceptually overlapping with interpretability findings here.

7) Transformers as interpretability proxies: Kriegeskorte’s recent work on transformer-based brain encoders as interpretable models is directly relevant to the proposed analysis.

I've been working in the field for years now and I perfectly understand that given the recent explosion of apporaches it's difficult to keep everything under track. Probably many other work could be cited here, but the ones I mentioned are particularly close to this work. Some of them are recent enough to be considered parallel concurrent work (for example a recently accepted paper at NeurIPS on interpretability on visual decoding with similar findings) but  I think integrating these citations and comparisons would better situate NeuroAdapter in the broader literature and clarify its novelty relative to ongoing work.

**Questions:**

- The metrics used are not particularly informative given current saturation; however, the authors appropriately acknowledge this. The emphasis on qualitative and interpretability results is appreciated and well justified.

- The examples are compelling, especially the attention visualizations and the demonstration of out-of-distribution (imagery) performance.

- The use of the encoder as a ranker is an elegant and practical idea; expanding on its generalization capabilities or possible integration as a training-time guidance signal could strengthen the paper.

That said, I do have some questions for the authors.

1) Could you elaborate on how the brain encoder–based ranking could be used during training (e.g., contrastive or self-distillation setups) rather than only at inference?

2) Have you considered test-time augmentation (e.g., sampling multiple seeds or perturbations of brain tokens) to better generalize to imagery data? The input distribution shift could be a key factor here. The model is working to some extent but lacks the precision of the visual task.

3) How does the interpretability of IBBI compare quantitatively to attribution methods such as SHAP or feature occlusion, maybe on yours or simpler models like Brain-Diffuser?

4) Why parcel based approach instead of voxel-wise one? I'm curios I've seen both approaches working fine and would like to better understand what is in your opinion the advantage of this strategy.

5) How do you expect NeuroAdapter to perform in a multi-subject joint training setting—does the architecture allow easy cross-subject fusion?

---

> ### Author Response · Authors · 2025-11-26
> **Response 1**
>
> We thank the reviewer for their thoughtful comments and the positive evaluation of our work! Please see below for our responses.
>
> > **Q1) Additional relevant literature**
>
> We agree with the reviewer and appreciate pointing out other relevant research in this fast-growing area. We have updated the manuscript to integrate all the suggested articles (listed below) into the relevant work.
>
> - Shinji Nishimoto, An T Vu, Thomas Naselaris, Yuval Benjamini, Bin Yu, and Jack L Gallant. Reconstructing visual experiences from brain activity evoked by natural movies. Current biology, 21(19):1641–1646, 2011.
>
> - Marlene Careil, Yohann Benchetrit, and Jean-Remi King. Dynadiff: Single-stage decoding of im- ages from continuously evolving fmri. arXiv preprint arXiv:2505.14556, 2025.
>
> - Matteo Ferrante, Tommaso Boccato, Furkan Ozcelik, Rufin VanRullen, and Nicola Toschi. Through their eyes: multi-subject brain decoding with simple alignment techniques. Imaging Neuroscience, 2:1–21, 2024.
>
> - Ethan Hwang, Hossein Adeli, Wenxuan Guo, Andrew Luo, and Nikolaus Kriegeskorte. In silico mapping of visual categorical selectivity across the whole brain. arXiv preprint arXiv:2510.21142, 2025.
>
> - Andrew Luo, Maggie Henderson, Leila Wehbe, and Michael Tarr. Brain diffusion for visual exploration: Cortical discovery using large scale generative models. Advances in Neural Information Processing Systems, 36:75740–75781, 2023a.
>
> - Andrew F Luo, Margaret M Henderson, Michael J Tarr, and Leila Wehbe. Brainscuba: Fine-grained natural language captions of visual cortex selectivity. arXiv preprint arXiv:2310.04420, 2023b.
>
> - Jerry Tang, Amanda LeBel, Shailee Jain, and Alexander G Huth. Semantic reconstruction of continuous language from non-invasive brain recordings. Nature Neuroscience, 26(5):858–866, 2023a.
>
> - Richard Antonello, Chandan Singh, Shailee Jain, Aliyah Hsu, Sihang Guo, Jianfeng Gao, Bin Yu, and Alexander Huth. Generative causal testing to bridge data-driven models and scientific theories in language neuroscience. arXiv preprint arXiv:2410.00812, 2024.
>
> - Vinamra Benara, Chandan Singh, John X Morris, Richard J Antonello, Ion Stoica, Alexander G Huth, and Jianfeng Gao. Crafting interpretable embeddings for language neuroscience by asking LLMs questions. Advances in neural information processing systems, 37:124137, 2024
>
> The following paper from Kriegeskorte lab (recently published in NeurIPS 2025) was already referenced in our paper since we used the encoding model architecture from that work. In our revision, we also discuss their relevance to the interpretability analyses.
>
> - Hossein Adeli, Minni Sun, and Nikolaus Kriegeskorte. Transformer brain encoders explain human high-level visual responses. arXiv preprint arXiv:2505.17329, 2025.

---

> ### Author Response · Authors · 2025-11-26
> **Response 2**
>
> > **Q2) Use of brain encoder prediction during training**
>
> We appreciate the suggestion. We plan to explore several ways to integrate the encoder into training, for example by optimizing joint decoding and encoding objectives. We also agree that the contrastive setting would be a promising direction to explore in the future work.
>
> > **Q3) Test-time augmentation**
>
> We perform test-time augmentations by generating multiple samples with different seeds and selecting the best reconstruction candidate according to our encoder. This process improves the overall performance (Appendix E, F, K).
>
> As for other perturbations, our experiments indicate that randomly masking tokens during test time generally leads to less accurate reconstructions. That said, we agree that a more systematic exploration of alternative perturbation schemes would be valuable, and we plan to investigate this in future work.
>
> > **Q4) Relation to other interpretability methods**
>
> We appreciate the reviewer’s suggestion to compare IBBI with attribution methods such as SHAP or feature occlusion. However, these approaches are fundamentally designed for feedforward models that map a static input to a static output. In contrast, our setting involves a multi-step generative process, where brain-derived tokens interact with the U-Net through cross-attention at each denoising timestep. SHAP and occlusion provide only static, input–output attributions and therefore cannot capture (i) how neural signals influence different diffusion steps, (ii) how contributions from specific ROIs evolve over time, or (iii) how brain information modulates spatial attention during image formation. This was our motivation to build decoding models that directly attend to the brain parcels.
>
>
> > **Q5) Parcel-wise vs voxel-wise**
>
> This is an excellent question. Models that use voxel-wise mapping usually aggregate them later in the pipeline to create a smaller number of tokens. Since spatially contiguous functional parcels already provide a biologically grounded local aggregation, we chose to use this approach. This choice also makes the model more lightweight (less memory and computation) with fewer number of parameters.
>
> > **Q6) Easy Multi-subject training?**
>
> We believe so. Multi-subject models often employ a subject specific linear mapping from neural data to tokens, followed by shared decoding components across subjects. Our architecture fits naturally into this multi-subject training recipe: we can introduce a separate linear mapping for each subject while sharing the downstream decoding modules. We plan to explore this in our future work.

---

> ### Comment · Reviewer_5Pr2 · 2025-11-26
> **Response to authors**
>
> I read author response and edited version of the paper. Where possible, they fully address my concerns. Otherwise they clearly reply with their intuition as in a good scientific debate.
>
> The work is overall solid, and I'm willing to continue to support acceptance.
> I'm raising my score to 8 to strenghen my support. I don't think the work is perfect, but I really do think that in science and especially in ICLR we should talk and discuss more about the topic and this is a nice paper that I would see there.
> Congrats and wish good luck to authors..

---

> > ### Author Response · Authors · 2025-11-28
> > **Response to Reviewer 5Pr2**
> >
> > We greatly appreciate your positive evaluation of our work and thank you again for your detailed feedback and engagement with our work!

---

### Official Review · Reviewer_hBgN · 2025-10-29

**Soundness:** 3
**Presentation:** 3
**Contribution:** 2
**Rating:** 6
**Confidence:** 3

**Summary:**

This paper introduces "NeuroAdapter," a framework for reconstructing visual images directly from fMRI brain activity. It bypasses the common two-stage approach of mapping brain signals to an intermediate feature space, instead conditioning a latent diffusion model directly on fMRI representations. The goal is to achieve competitive image reconstruction quality while also improving interpretability. The authors also propose an "Image-Brain BI-directional interpretability framework" (IBBI) that analyzes cross-attention mechanisms to visualize how different brain regions contribute to the image generation process.

**Strengths:**

This paper proposes a framework that directly conditions a generative model on fMRI data, bypassing the intermediate feature mapping common in two-stage methods. A primary contribution is the IBBI interpretability framework, which is designed to analyze cross-attention weights to visualize how specific brain regions influence the image generation. The method also demonstrates competitive decoding performance, especially on semantic metrics, and shows generalization capabilities to mental imagery tasks and novel image categories.

**Weaknesses:**

- The model uses a "Parcel-wise Linear Mapping" to transform fMRI data from each brain parcel into an embedding. This is a strong simplification. It assumes that the complex, high-dimensional neural information within an entire parcel can be sufficiently captured by a single linear projection. This could limit the amount of neural detail the model can use.

- The methodology section notes that for evaluation, multiple images are generated, and a separate "Brain Encoder" model is used to select the best one. This complicates the claim of an "end-to-end" framework, as the final, high-quality output relies on a post-processing selection step that is external to the generative model itself.

- The paper itself notes "modest" performance on low-level metrics (like pixel correlation and SSIM). It justifies this by stating it intentionally omitted a separate pathway for low-level features (which other models use) to favor interpretability. While this is a reasonable design choice, the model sacrifices high-fidelity, pixel-accurate visual reconstruction for semantic coherence and interpretability, which is a key weakness.

**Questions:**

- The methodology relies on a separate "Brain Encoder" to select the best reconstruction from a batch of generated candidates. How dependent are the reported "competitive" performance metrics on this selection process? What is the average quality of a randomly selected (or "raw") generated image from the model, and how does this compare to the "best" selected image?

- The model maps the complex, high-dimensional fMRI activity from an entire brain parcel to a single embedding using a simple linear projection. Is this linear mapping truly sufficient to capture the rich information in that parcel, or does it serve as a significant information bottleneck? Were more complex, non-linear mappings (e.g., small MLPs per parcel) explored, and if so, how did they affect performance and training stability?

---

> ### Author Response · Authors · 2025-11-26
> **Response**
>
> We are thankful to reviewer for the positive evaluation and the constructive review. Please see below for our responses to the questions.
>
> > **Q1) Brain encoder effect**
>
> Great suggestion. Due to the stochastic nature of the diffusion models, the generative process can vary across sampling seeds. We used the brain encoder as a post-hoc mechanism to "stabilize" decoding by selecting the candidate whose predicted neural response best matches the measured fMRI. As shown in the table in **Appendix K** (copied below), increasing the number of predictions with different random seeds provides consistent improvement in performance.
>
> To address the reviewer's comment, we have performed an additional experiment to further understand the role of the brain encoder in decoding. For each test sample, we generated 8 reconstructions using 8 different random initializations and saved all candidates. We then computed image-quality metrics under three conditions: the Highest-Corr image selected by the brain encoder among candidates, the Lowest-Corr image, and a Random image sampled uniformly from the eight candidates. As shown in the table below, the brain-encoder selection improves performance across most metrics when compared to the Lowest-Corr and Random choices. However, we also observe that randomly selected images sometimes outperform the Highest-Corr images on certain metrics. Using our brain encoder for ranking does not directly optimize perceptual metrics, but instead selects the reconstruction that is most aligned with the neural data.
>
> Table in Appendix K
> | Num_Preds | PixCorr↑ | SSIM↑ | AN(2)↑ | AN(5)↑ | Incep↑ | CLIP↑ | Eff↓  | SwAV↓ |
> |-----------------|----------|-------|--------|--------|--------|-------|-------|-------|
> | 1            | .104    | .292 | 79.0%   | 90.1%   | 89.5%  | 90.8%  | .729  | .417   |
> | 2            | .105    | .292 | 82.6%   | 91.7%   | 89.8%  | 88.7%  | .733  | .416   |
> | 4            | .120    | .293 | 84.0%   | 93.5%   | 90.1%  | 91.5%  | .725  | .408   |
> | 8            | **.133**| **.296** | **86.22%** | **93.96%** | **92.15%** | **92.03%** | **.706** | **.396** |
>
> Table for new experiment
> | Condition | PixCorr↑ | SSIM↑ | AN(2)↑ | AN(5)↑ | Incep↑ | CLIP↑ | Eff↓  | SwAV↓ |
> |-----------------|----------|-------|--------|--------|--------|-------|-------|-------|
> | Lowest Corr  | 0.109     | **0.286** | 80.88%  | 89.51%  | 89.81%  | 91.31% | 0.720  | 0.404  |
> | Random  | 0.130     | **0.286**  | 84.00%  | 91.30%  | **92.05%**  | 92.33% | 0.708  | 0.395  |
> | Highest Corr  | **0.139**     | 0.285  | **88.02%**  | **93.87%**  | 91.91%  | **93.01%** | **0.702**  | **0.387**  |
>
>
> > **Q2) Linear projection**
>
> The linear mapping is usually used to transform brain data to tokens and we made sure that it is not restrictive in this case. We used a fine-grained Schaefer parcellation (∼400–600 vertices per parcel), and each parcel is projected into a 768-dimensional embedding space, a dimension slightly larger than the number of vertices per parcel, ensuring that **the mapping is not compressive and does not discard representational degrees of freedom**. We also confirmed that the brain activity can be reconstructed from the tokens with near zero mean squared error loss. As shown in the table below (also in **Appendix H**) in our previous submission, we performed an ablation on the projection dimension. The results align with your intuition that projecting parcels into lower-dimension spaces creates an information bottleneck, decreasing the performance on low-level image-quality metrics.
>
> Additionally, an important goal of our framework is to make the model training lightweight, so the trainable module in our pipeline has **only 106.54M trainable parameters**. Using separate multi-layer perceptrons for all 200 parcels would substantially increase memory and computational cost. Nevertheless, we agree that non-linear parcel-wise mappings is a promising direction for future work.
>
> Table in Appendix H
> | Dim | PixCorr | SSIM | Alex(2) | Alex(5) | Incep | CLIP | Eff | SwAV |
> |----|---------|-------|---------|---------|--------|--------|-------|--------|
> | 1024 | .116 | **.303** | 81.49% | 91.64% | 88.17% | 90.33% | .742 | .432 |
> | 960  | .134 | .302 | 85.49% | 93.38% | 91.32% | 90.62% | .720 | .410 |
> | 768  | .133 | .296 | 86.22% | 93.96% | **92.15%** | **92.03%** | **.706** | **.396** |
> | 576  | .132 | .297 | **86.31%** | 93.87% | 90.42% | 90.76% | .712 | .400 |
> | 384  | **.136** | .301 | 85.65% | 94.16% | 89.78% | 90.13% | .725 | .411 |
> | 192  | .122 | .290 | 85.25% | **94.17%** | 91.33% | 90.78% | .718 | .417 |
>
>
> We have updated the paper to reflect these points.

---

### Official Review · Reviewer_SDwm · 2025-10-30

**Soundness:** 2
**Presentation:** 2
**Contribution:** 2
**Rating:** 4
**Confidence:** 2

**Summary:**

The paper proposes NeuroAdapter, an end‑to‑end brain‑to‑image decoding framework that directly conditions a latent diffusion model on parcel‑wise fMRI tokens, bypassing intermediate latent feature spaces. It introduces IBBI (Image–Brain BI‑directional Interpretability) to analyze cross‑attention across diffusion timesteps, offering a brain‑directed view (parcel contributions) and an image‑directed view (ROI attention maps). Experiments on NSD, NSD‑Imagery, and Deeprecon show competitive performance on high‑level semantic metrics relative to embedding‑aligned baselines, along with extensive ablations. Code and training details are provided for reproducibility.

**Strengths:**

- NeuroAdapter replaces U‑Net cross‑attention with an IP‑Adapter–style module and feeds only fMRI tokens. SD weights are frozen, and only the parcel‑wise linear mapper and new cross‑attention layers are trained. This separation makes the conditioning path easy to analyze and replicates the setup across datasets.

- Results are reported on NSD, NSD‑Imagery, and Deeprecon, helping assess generalization beyond the main dataset. Qualitative grids and quantitative tables are provided in the main text and appendices.

- The paper also analyzes cross‑attention dynamics. It reveals the contribution of attention at the pixel level throughout the generative process. It also uses parcel‑wise attention to quantify contributions at the image level. (That said, it is unclear whether these analyses yield any genuinely new insights beyond what was previously known. Please see weaknesses and questions.)

- Ablations cover fMRI token dropout, condition dimension, # of high‑SNR parcels, etc.

- The paper provides code, hardware/training schedules, and describes dataset preprocessing and metrics.

**Weaknesses:**

- While high‑level metrics are competitive, PixCorr/SSIM are below strong baselines (e.g., Brain Diffuser, MindEye1).

- The ROI attention maps (Face/Body/Scene/Word) and parcel contribution maps mostly illustrate expected effects (e.g., category ROIs attend to corresponding image regions). The paper does not yet demonstrate novel neuroscientific insights uniquely enabled by IBBI.

- If the emphasis is understanding brain–feature correspondences, a conventional encoding framework (predicting brain activity from controlled features) is arguably the more direct tool. The manuscript does not fully justify why decoding is the right or necessary vehicle for the questions posed. For example, consider how “face‑selective” cortex was originally identified: via encoding analyses that directly linked face features to measured brain responses. In the present paper, the ROIs used for interpretability (Face/Body/Scene/Word) are not discovered by their decoding models; they are imported via a label‑mapping heuristic, and the ROI‑attention visualizations and ROI‑masking largely reaffirm known selectivities rather than revealing new ones. This leaves unclear what scientific insight requires the decoding step (as opposed to a targeted encoding analysis).

**Questions:**

- Beyond confirming known category selectivity, what new neuroscientific findings does IBBI enable? Can you formalize and test such hypotheses with statistics?

- Given your use of a brain encoder and the standard role of encoding in neuroscience, please justify why decoding is preferable for the interpretability aims here—what questions can IBBI answer that a well‑designed encoding analysis could not?

---

> ### Author Response · Authors · 2025-11-26
> **Response**
>
> We thank the reviewer for the thoughtful review. Please see below for our responses.
>
> > **Q1) Performance on low-level meterics**
>
> Our main goal is not to compete for state-of-the-art performance on image quality metrics. Many of these metrics **are already close to saturation in the NSD setting**. Our reconstructions still capture the perceptually meaningful structure of the stimuli, as shown in the qualitative examples. Further boosting pixel-level details to improve metric results is certainly possible, such as using latest generative backbone to output high-resolution images, but we believe such improvements may not yield deeper insight. Instead, we focused on understanding how brain data is interacted with and utilized in the decoding process.
>
> In addition, our work also evaluates models on NSD imagery and Deeprecon datasets and has demonstrated that **our approach shows promising generalization capabilities across both subjects and task conditions**. However, many existing decoding methods are evaluated on NSD alone, including all models we compared against, making it difficult to assess whether their improvements reflect great generalization or dataset-specific overfitting. Our cross-dataset results therefore provide a stronger indication of robustness and supports the broader applicability of our framework.
>
> > **Q2) Scientific discovery from decoding analyses**
>
> We share the reviewer’s interest in the neuroscientific implications of our framework. We would like to clarify that the primary purpose of IBBI is not to explain brain computation directly, but to make the decoding process itself transparent. In other words, we aim to understand how brain-derived information is incorporated into the denoising trajectory of a diffusion model. Brain decoding has become important in brain-computer interface because it provides a way to read out information from neural activity, and in this context, we argue that a decoding model should not become overly complex or pursue marginal performance gains at the expense of transparency. The nature of the cross-attention mechanism in our architecture allows IBBI to serve as a diagnostic tool. It gives us a new perspective to see how the model learns to interact with brain signals to generate visual content step by step.
>
>
> A central scientific motivation for using decoding in addition to encoding analyses is that they address complementary questions. Encoding models characterize how external stimuli are transformed into neural responses. Decoding instead asks what aspects of visual or mental content can be reliably read out from measured activity, which is particularly important in settings where the relevant subjective percept is only partially constrained or cannot be fully specified by an external stimulus (e.g., visual illusion [1], mental imagery [2], dreams [3], and other forms of subjective perception). To leverage decoding for scientific insight, it is essential to understand how a decoding model uses brain signals to guide image generation. Existing analyses of latent diffusion models examine when (diffusion time step) and where (model layer) low- and high-level features emerge (Takagi & Nishimoto, 2023), but typically lack a time-resolved view of which parts of the generated image are modulated by brain-derived information. Our work addresses this gap by using cross-attention to provide explicit, temporally resolved maps linking brain signals to image regions throughout the denoising process.
>
> We have revised the paper to make the motivation more clear and hope this addresses the reviewer's concern. We welcome further discussion.
>
> [1] Fan L Cheng, Tomoyasu Horikawa, Kei Majima, Misato Tanaka, Mohamed Abdelhack, Shuntaro C Aoki, Jin Hirano, and Yukiyasu Kamitani. Reconstructing visual illusory experiences from human brain activity. Science Advances, 9(46):eadj3906, 2023.
>
> [2] Reese Kneeland, Paul S. Scotti, Ghislain St-Yves, Jesse Breedlove, Kendrick Kay, and Thomas Naselaris. Nsd-imagery: A benchmark dataset for extending fmri vision decoding methods to mental imagery. In 2025 IEEE/CVF Conference on Computer Vision and Pattern Recognition (CVPR), pp. 28852–28862. IEEE, June 2025. doi: 10.1109/cvpr52734.2025.02687. URL http: //dx.doi.org/10.1109/CVPR52734.2025.02687.
>
> [3] Horikawa, T., Tamaki, M., Miyawaki, Y., & Kamitani, Y. (2013). Neural decoding of visual imagery during sleep. Science (New York, N.Y.), 340(6132), 639–642. https://doi.org/10.1126/science.1234330

---

### Official Review · Reviewer_rWiu · 2025-11-04

**Soundness:** 3
**Presentation:** 3
**Contribution:** 2
**Rating:** 4
**Confidence:** 3

**Summary:**

This paper introduces NeuroAdapter, an end-to-end framework for brain-to-image decoding that directly conditions a latent diffusion model on fMRI-derived parcel embeddings, bypassing intermediate image/text feature spaces. The key technical idea is to inject parcel-wise brain tokens into Stable Diffusion via an IP-Adapter–style cross-attention module while keeping the text encoder empty. To probe how brain signals guide generation, the authors propose IBBI framework, which analyzes cross-attention weights over diffusion steps from two complementary views: (i) a brain-directed view that quantifies parcel/ROI contributions over time, and (ii) an image-directed view that produces ROI attention maps highlighting where cortical regions attend in the generated image. Experiments on three datasets show that NeuroAdapter attains competitive reconstruction performance (especially on high-level semantic metrics) and provides intuitive interpretability visualizations.

**Strengths:**

- Contribution to interpretability is clear. Interpretability is important for both AI and neuroscience. The IBBI framework gives a principled way to link parcel-level signals to image-space attention at each diffusion step, which improves transparency of the decoding pipeline.

- Interesting use of a brain encoder as a ranking tool. Using a separately trained brain encoder to score multiple decoded candidates by their predicted brain responses is a simple, effective mechanism to pick the final reconstruction.

- Thorough experiments on three datasets. The paper evaluates on NSD (with multiple subjects), NSD-Imagery, and Deeprecon. The metric suite includes both low-level (e.g., PixCorr, SSIM) and high-level (e.g., AlexNet/CLIP/Inception-based) measures, with ablations on token dropout, parcel count, and condition dimension.

- Comprehensive appendix. The supplementary materials provide substantial details (data processing, model choices, ablations, additional visualizations), which improve clarity and reproducibility.

**Weaknesses:**

- Interpretability claims rely primarily on qualitative evidence. Although Figures highlighting ROI contributions and attention maps are compelling, the paper does not rigorously quantify interpretability (e.g., ROI-level correspondence metrics, sensitivity/perturbation scores summarized across subjects/ROIs, or agreement with independent neuroscience priors/experts). To make the interpretability claim more convincing, the authors are suggested to report a quantitative metric for benchmarking interpretability specifically.
- With proper metrics defined for interpretability, the authors are suggested to compare NeuroAdapter’s outputs directly to those from related attention-map–producing baselines to confirm the effectivess of the proposed approach on interpretability.

- Compute/efficiency and model size are unclear. Because NeuroAdapter adds an IP-Adapter–style conditioning path and uses 50 denoising steps with candidate sampling before brain-encoder ranking, it is important to clarify whether inference time and parameter count remain comparable to competing methods. Given that the main tables/figures show NeuroAdapter is competitive on high-level metrics but not leading on low-level metrics, the method could be at a disadvantage if it is substantially larger or slower.

- Moreover, it is unclear when reporting results in Figure 5 or Table 1, the brain encoder ranking tool was used or not. The method is interesting but pretty stand-alone to the NeuroAdapter itself. Other methods may benefit from the design too.

**Questions:**

Please see the weakness section.

---

> ### Author Response · Authors · 2025-11-26
> **Response 1**
>
> We really appreciate the thoughtful and detailed review! Below we respond to the comments and questions.
>
>
> > **Q1) Quantifying the interpretability results**
>
> Great suggestion. To quantify the interpretability results, we evaluate ROI-specific IBBI attention maps using Intersection-over-Union (IoU) and Dice scores, which measure the spatial overlap between predicted attention regions and semantic segmentation masks from the latest Segment Anything 3 model (SAM3; [1]). SAM3 provides high-quality region segmentation and serves as pseudo–ground-truth for our generated images. Among the 515 test reconstructions, 236 images contain valid semantic regions, including 38 Face, 195 Body, 27 Scene, and 7 Word images.
>
> For IBBI masks, each ROI produces a 2D attention map over denoising steps. We average attention across steps, normalize, and apply a 50% threshold to obtain binary masks representing ROI-specific attended regions, following the DAAM procedure (Zhang et al., 2023). As a baseline, we use a whole-image mask, representing the strategy of “attending everywhere.”
>
> We compute IoU and Dice between predicted masks and SAM3 masks. Face, Body, and Word ROIs show substantially higher IoU and Dice scores with IBBI attention maps compared to the whole-image baseline, demonstrating that IBBI reliably localizes semantically meaningful regions. The only exception is the scene ROIs: the scene masks returned by SAM3 typically cover large, contiguous background regions, which inflates IoU/Dice for the whole-image baseline because most pixels belong to the “scene” class. In contrast, IBBI allocates attention selectively to relevant subregions rather than spreading uniformly across the entire background.
>
> | IoU | Face | Body | Scene | Word |
> |---------|------|------|-------|------|
> | Whole-image Mask| 0.124 | 0.171  | 0.605  | 0.199  |
> | IBBI Attn Mask   | 0.322 | 0.362  | 0.198  | 0.453  |
>
> | Dice | Face | Body | Scene | Word |
> |---------|------|------|-------|------|
> | Whole-image Mask | 0.213 | 0.276  | 0.717  | 0.313  |
> | IBBI Attn Mask    | 0.459 | 0.504  | 0.314  | 0.595  |
>
> We have added these new analyses to the revised paper.
>
> [1] Carion, Nicolas, Laura Gustafson, Yuan-Ting Hu, Shoubhik Debnath, Ronghang Hu, Didac Suris, Chaitanya Ryali, et al. 2025. “SAM 3: Segment Anything with Concepts” (arXiv).
>
>
>
> > **Q2) Comparison to other attention methods**
>
> To our knowledge, our work is the first to be able to create attention maps showing which brain areas were attended, to generate different parts of the image. This was our main motivation to build a model without the intermediate mapping that obscures these effects. The two-stage methods typically map the whole brain to the intermediate representation that loses area specific information.

---

> ### Author Response · Authors · 2025-11-26
> **Response 2**
>
> > **Q3) Compute efficiency and model size**
>
> Thank you for raising this point. In our framework, the parcel-wise linear mapping module has 86.78M parameters and the Cross-attention module has 19.76M parameters (constituting only 9.08% of total parameters). This is substantially smaller than typical two-stage decoding approaches, which often have on the order of 1 billion parameters (e.g. MindEye). Creating a lightweight decoding alternative was part of the motivation to build NeuroAdapter.
>
> We used 50 inference steps, following the default setting in Stable Diffusion. In response to the reviewer's comment, we varied the number of denoising steps and observed similarly good decoding performance around 50 steps (see table below).  We added these results in the revision (Appendix L).
>
> Table for new experiment
> | Denoising Steps | PixCorr↑ | SSIM↑ | AN(2)↑ | AN(5)↑ | Incep↑ | CLIP↑ | Eff↓  | SwAV↓ |
> |-----------------|----------|-------|--------|--------|--------|-------|-------|-------|
> | 20  | 0.137     | 0.291  | 87.92%  | 95.46%  | 90.49%  | 90.34% | 0.707  | 0.396  |
> | 30  | 0.135     | 0.291  | 87.93%  | 94.50%  | 90.90%  | 90.93% | 0.706  | 0.391  |
> | 40  | 0.142     | 0.299  | 88.06%  | 95.96%  | 91.58%  | 91.83% | 0.705  | 0.388  |
> | 50  | 0.143     | 0.289  | 87.96%  | 95.42%  | 91.72%  | 91.19% | 0.705  | 0.389  |
> | 60  | 0.141     | 0.288  | 87.26%  | 94.61%  | 91.50%  | 91.62% | 0.702  | 0.386  |
> | 70  | 0.137     | 0.284  | 87.40%  | 94.85%  | 91.46%  | 91.10% | 0.700  | 0.389  |
> | 80  | 0.138     | 0.285  | 87.13%  | 94.47%  | 90.77%  | 90.43% | 0.704  | 0.389  |
>
>
>
> > **Q4) Clarification on the use of brain encoder for ranking**
>
> All reported results in Figure 5 and Table 1 are obtained with the brain-encoder ranking step, because this is part of our standard inference pipeline. For completeness, we report performance on Subject 1 without brain-encoder ranking in **Appendix K** (copied below). We view this as **a general plug-and-play component** for the post-hoc decoding step. Importantly, our encoder-based selection step does not alter training or give our method an unfair advantage. Prior work also adopted candidate image selection procedures due to the inherent noise in the neural data and the stochasticity of the diffusion models. For example, Takagi et al. (2023, section 3.3) generated 5 images for each test image and selected the generated images with highest perceptual similarity metrics; MindEye1 (2023, section 3.2) selected the best of 16 reconstructions by ranking candidates with CLIP similarity to brain-predicted embeddings; MindBridge (2024, Appendix B) selected one from 8 reconstructions with the highest CLIP similarity to the stimuli as the final result. Notably, our ranking is driven solely by how well each candidate predicts the measured neural activity, ensuring that **the final reconstruction remains faithful to the brain data**.
>
> Table in Appendix K
> | Num_Preds | PixCorr↑ | SSIM↑ | AN(2)↑ | AN(5)↑ | Incep↑ | CLIP↑ | Eff↓  | SwAV↓ |
> |-----------------|----------|-------|--------|--------|--------|-------|-------|-------|
> | 1            | .104    | .292 | 79.0%   | 90.1%   | 89.5%  | 90.8%  | .729  | .417   |
> | 2            | .105    | .292 | 82.6%   | 91.7%   | 89.8%  | 88.7%  | .733  | .416   |
> | 4            | .120    | .293 | 84.0%   | 93.5%   | 90.1%  | 91.5%  | .725  | .408   |
> | 8            | **.133**| **.296** | **86.22%** | **93.96%** | **92.15%** | **92.03%** | **.706** | **.396** |
>
> We have revised the paper to clarify these points.

---

### Author Response · Authors · 2025-12-01
**General response**

We are deeply grateful for the reviewers' detailed and constructive comments and thank the AC for their time and for overseeing our submission.

> **Paper Summary**

Our work presents NeuroAdapter, a visual decoding framework that learns parcel-wise embeddings directly from fMRI data and integrates them into a latent diffusion model through cross-attention. Our approach has been evaluated on 3 different public fMRI datasets, showing that high-quality visual reconstructions can be obtained without relying on external image or text embedding spaces. In addition, we introduce IBBI, which analyzes cross-attention dynamics across diffusion steps, making the decoding pipeline more transparent.

> **Review Summary**

We greatly appreciate that all four reviewers highlighted the different strengths of our work:
- Offering comprehensive evaluation across three datasets, demonstrating generalization beyond NSD (**rWiu**, **SDwm**, **hBgN**, **5Pr2**).
- Achieving competitive decoding performance on high-level semantic metrics, even without intermediate embedding spaces (**hBgN**，**5Pr2**).
- Validating the effectiveness of the NeuroAdapter design through extensive ablation studies (**rWiu**, **SDwm**, **5Pr2**).
- Demonstrating clear contributions to interpretability via the IBBI framework (**rWiu**, **hBgN**, **5Pr2**).
- Providing high-quality paper presentation and comprehensive appendices (**rWiu**, **SDwm**, **5Pr2**)


Summarized below are clarifications and additional analyses in the revised paper in response to reviewers' feedback.

- We added discussions of several other papers in this fast-growing field to the relevant works section. (**5Pr2**)

- We further expanded on how IBBI is designed to make the decoding process itself transparent. In other words, we aim to understand how brain-derived information is incorporated into the denoising trajectory of a diffusion model (**SDwm**). As reviewer **5Pr2** pointed out, "the metrics are kind of saturated" on the NSD dataset. We believe the decoding pipeline should not become overly complex to pursue marginal performance gains at the expense of transparency. Instead, we focused on understanding how brain data is utilized in the decoding process. The nature of the cross-attention mechanism in our architecture allows IBBI to serve as a diagnostic tool. It gives us a new perspective on how the model learns to interact with brain signals to generate visual content step by step. (**rWiu**, **SDwm**, **hBgN**)

- We added further analyses to quantify the IBBI attention maps. We compute IoU and Dice between predicted masks and SAM3 segmentation masks. Face, Body, and Word ROIs show substantially higher IoU and Dice scores with IBBI attention maps compared to the whole-image baseline (**Table 1** and **Appendix O**), demonstrating that IBBI reliably localizes semantically meaningful regions. (**rWiu**)


- We also added more details on model size and compute efficiency compared to alternative approaches, and verified the default choice of 50 denoising steps is near optimal in our setup (**Appendix L**). (**rWiu**)


- We provided further analyses of how brain encoder ranking of the generations impacts the overall results, showing how the metrics would change if a generation is randomly selected vs based on the encoder predictions. **Appendix K (Table 12)**.(**rWiu**, **hBgN**)


- We clarified how the choice of token dimension impacts the reconstruction accuracy, and showed that the choice of 768 is generally optimal. (**Appendix H**) (**hBgN**)

> **Conclusion**

We sincerely thank all reviewers for the constructive and insightful feedback. We believe the requested clarifications and additional experiments significantly improved the communication of our work.

We look forward to further discussion and are happy to address any additional questions!

---

### Meta-Review · Area_Chair_CfyF · 2026-01-12

**Summary:**

This paper proposes NeuroAdapter, an end-to-end brain-to-image decoding framework that directly conditions a latent diffusion model on parcel-wise fMRI representations, bypassing intermediate image/text embedding spaces. A central contribution is the IBBI interpretability framework, which analyzes cross-attention across diffusion timesteps to reveal how different cortical regions influence image generation.

Across four reviews, there is broad agreement that the paper is technically sound, well presented, and thoughtfully executed, with strengths in interpretability, architectural clarity, and multi-dataset evaluation (NSD, NSD-Imagery, DeepRecon). Reviewers particularly valued the transparency enabled by cross-attention to brain parcels, the comprehensive ablations, and the effort to move beyond pure reconstruction metrics.

**Reviewer Concerns:**

Concerns largely addressed:

- Lack of quantitative interpretability evidence.
- Unclear compute cost and model size.
- Dependence on brain-encoder ranking.
- Missing or insufficient related work.

Concerns partially outstanding:
- Limited low-level (pixel/SSIM) performance relative to best baselines.
- Neuroscientific novelty.

**Reviewer Scores:**

Two are postive, and two are borderline towards postive.

---

### Decision · Program_Chairs · 2026-01-26

Accept (Poster)